# Generating Full-field Evolution of Physical Dynamics from Irregular Sparse Observations

**Panqi Chen**[1]  **Yifan Sun**[1]  **Lei Cheng**[1,2] *  **Yang Yang**[3]  **Weichang Li**[1]
**Yang Liu**[4]  **Weiqing Liu**[4]  **Jiang Bian**[4]  **Shikai Fang**[1,4] *

[1] College of Information Science and Electronic Engineering, Zhejiang University
[2] Zhejiang Provincial Key Laboratory of
Multi-Modal Communication Networks and Intelligent Information Processing
[3] College of Computer Science and Technology, Zhejiang University
[4] Microsoft Research Asia

## Abstract

Modeling and reconstructing multidimensional physical dynamics from sparse and off-grid observations presents a fundamental challenge in scientific research. Recently, diffusion-based generative modeling shows promising potential for physical simulation. However, current approaches typically operate on on-grid data with preset spatiotemporal resolution, but struggle with the sparsely observed and continuous nature of real-world physical dynamics. To fill the gaps, we present SDIFT, **S**equential **DI**ffusion in **F**unctional **T**ucker space, a novel framework that generates full-field evolution of physical dynamics from irregular sparse observations. SDIFT leverages the functional Tucker model as the latent space representer with proven universal approximation property, and represents observations as latent functions and Tucker core sequences. We then construct a sequential diffusion model with temporally augmented UNet in the functional Tucker space, denoising noise drawn from a Gaussian process to generate the sequence of core tensors. At the posterior sampling stage, we propose a Message-Passing Posterior Sampling mechanism, enabling conditional generation of the entire sequence guided by observations at limited time steps. We validate SDIFT on three physical systems spanning astronomical (supernova explosions, light-year scale), environmental (ocean sound speed fields, kilometer scale), and molecular (organic liquid, millimeter scale) domains, demonstrating significant improvements in both reconstruction accuracy and computational efficiency compared to state-of-the-art approaches. The code is available at `https://github.com/OceanSTARLab/SDIFT`.

## 1 Introduction

Modeling and reconstructing spatiotemporal physical dynamics in the real world has been a long-standing challenge in science and engineering. In fields such as aeronautics, oceanography, and climate simulation, we often need to infer the evolution of physical dynamics in continuous space-time from sparse observations. For example, meteorologists must predict weather patterns from limited station data, while ocean scientists reconstruct underwater current dynamics from sparse sensors. Traditional numerical methods have been widely applied, but they typically require substantial computational resources and struggle to ensure fine-scale prediction in complex scenarios.

Recently, generative models – particularly diffusion models [1, 2, 3, 4, 5, 6] – have achieved remarkable success in computer vision and related fields. These models have demonstrated strong capa-

---

*Correspond to lei_cheng@zju.edu.cn, fsk@zju.edu.cn

bilities in generating high-quality images, videos, and other complex data structures. An increasing number of works have begun to explore the use of generative models for physical simulation, such as modeling underwater sound speed, turbulence, and other complex phenomena, addressing tasks like PDE solving, super-resolution, and reconstruction[4, 5, 6].

A typical paradigm in physical diffusion modeling involves constructing an autoencoder-like mapping from the observation space to a latent space, using visual feature extractors such as convolutional neural networks[7] (CNNs) or Vision Transformers[8] (ViTs) as encoder/decoder architectures. A diffusion model is then trained in the latent space. During inference/sampling, diffusion models use the likelihood of observations as guidance and generate target dynamics via diffusion posterior sampling[9] (DPS), which are then mapped back to the observation space. While this paradigm has shown success, it also faces several limitations in real-world physical systems. For instance, physical observations are typically obtained from sparse sensors that sample irregularly across space and time, and are often corrupted by noise – conditions under which visual-based encoders are not suitable. Moreover, the goal of posterior sampling is to generate the full-field evolution of dynamics, including inference at arbitrary continuous spatial/temporal coordinates, yet observations are only available at a few discrete time steps – making standard DPS ineffective.

To bridge these critical gaps, we introduce SDIFT (Sequential Diffusion in Functional Tucker Space), a generative framework that reconstructs continuous physical dynamics from sparse, irregular observations. SDIFT uses a Functional Tucker Model (FTM) to encode data into structured latent functions and core tensors. We prove that the FTM universally approximates complex, multidimensional physical systems. Building upon this latent representation, SDIFT applies Gaussian Processbased Sequential Diffusion (GPSD) to produce temporally coherent core sequences at arbitrary time resolutions. Crucially, we propose a novel Message-Passing Diffusion Posterior Sampling (MPDPS) mechanism that enhances conditional generation capabilities, effectively propagating observation-driven guidance across the entire spatiotemporal sequenceeven at unobserved timesteps. We evaluate SDIFT on three diverse physical domainssupernova explosions, ocean sound speed, and active particle systemsand demonstrate its superior accuracy, noise robustness, and computational efficiency compared to stateoftheart methods, even under extreme sparsity and irregular sampling, underscoring its broad applicability and practical relevance.

Our contributions can be succinctly summarized as follows: 1) We propose SDIFT, a novel generative framework designed to reconstruct multidimensional physical dynamics from sparse, irregularly observed data, significantly advancing current methodologies. 2) We establish the theoretical foundation of the FTM as a universal latent representer and integrate it within a Gaussian process-based diffusion model to achieve robust and flexible generation of latent core sequences. 3) We introduce MPDPS, an innovative posterior sampling method, facilitating effective message propagation from sparse observations, ensuring robust reconstruction across temporal domains. 4) We demonstrate state-of-the-art performance across three distinct physical phenomena at various scales, highlighting the framework's generalizability and robustness. This work paves the way for more flexible, accurate and computationally efficient reconstructions of real-world physical systems, with broad implications for simulation and prediction in complex scientific and engineering tasks.

## 2 Preliminary

### 2.1 Tensor Decomposition

**Standard tensor decomposition** operates on a $K$-mode tensor $\mathcal{Y} \in \mathbb{R}^{I_1 \times \cdots \times I_K}$, where the $k$-th mode has $I_k$ nodes, and each entry $y_\mathbf{i}$ is indexed by a $K$-tuple $\mathbf{i} = (i_1, \cdots, i_k, \cdots, i_K)$, where $i_k$ denotes the index of the node along the mode $k$. The classical Tucker decomposition[10] utilizes a group of latent factors $\{\mathbf{U}^k \in \mathbb{R}^{I_k \times R_k}\}_{k=1}^K$ and a core tensor $\mathcal{W} \in \mathbb{R}^{R_1 \times \cdots \times R_k \times \cdots \times R_K}$, known as the Tucker core tensor, to formulate a compact representation of the tensor, where $\{R_k\}_{k=1}^K$ is the preset rank for latent factors in each mode. Then, each entry is modeled as the multi-linear interaction of involved latent factors and the core tensor:

$$y_\mathbf{i} \approx \text{vec}(\mathcal{W})^\mathrm{T}(\mathbf{u}_{i_1}^1 \otimes \cdots \otimes \mathbf{u}_{i_k}^k \otimes \cdots \otimes \mathbf{u}_{i_K}^K) = \sum_{r_1}^{R_1} \cdots \sum_{r_K}^{R_K} [w_{r_1, \cdots, r_K} \prod_{k=1}^K u_{i_k, r_k}^k], \quad (1)$$

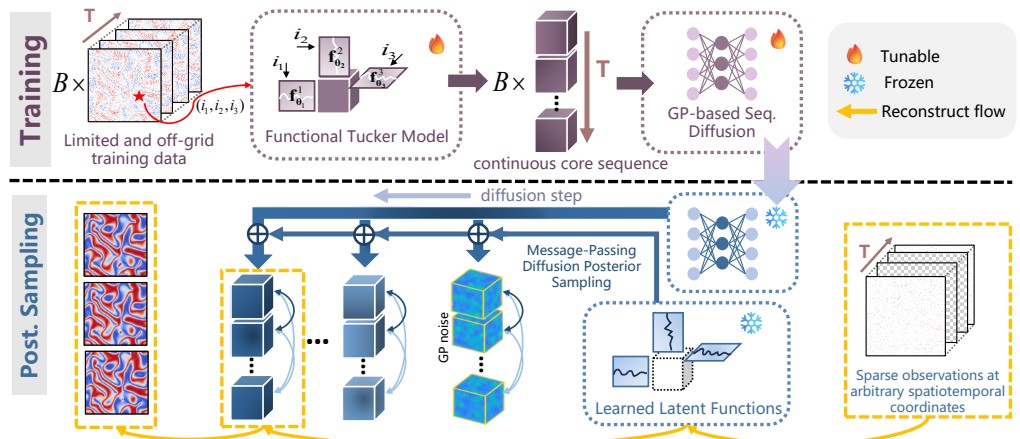

Figure 1: Graphical illustration of the proposed SDIFT with MPDPS mechanism.

where $\mathbf{u}_{i_k}^k \in \mathbb{R}^{R_k}$ is the $i_k$-th column of $\mathbf{U}^k$, corresponding to the latent factor of node $i_k$ in mode $k$, $\text{vec}(\cdot)$ is the vectorization operator, and $\otimes$ is the Kronecker product. $w_{r_1, \cdots, r_K}$ is the $(r_1, \cdots, r_K)$-th element of $\boldsymbol{\mathcal{W}}$ and $u_{i_k, r_k}^k$ is the $r_k$-th element of $\mathbf{u}^k(i_k)$.

**Functional tensor** is the generalization of tensor to functional domain. Standard tensors work with data defined on fixed grids, where each mode has a finite number of nodes and entries are indexed discretely. In contrast, functional tensors view the entries as samples from a multivariate function defined over continuous domains. This function can be decomposed into mode-wise components. Using the Tucker format, this idea leads to the functional Tucker decomposition [11, 12], which is expressed as:

$$y(\mathbf{i}) \approx \text{vec}(\boldsymbol{\mathcal{W}})^{\mathrm{T}}(\mathbf{u}^1(i_1) \otimes \cdots \otimes \mathbf{u}^k(i_k) \otimes \cdots \otimes \mathbf{u}^K(i_K)), \tag{2}$$

where $\mathbf{i} = (i_1, \cdots, i_K) \in \mathbb{R}^K$ denotes the index tuple, and $\mathbf{u}^k(\cdot) : \mathbb{R} \to \mathbb{R}^{R_k}$ is the latent vector-valued function of mode $k$. The continuous nature of functional tensor makes it suitable for many real-world applications with high-dimensional and real-valued coordinates, like the climate and geographic data.

## 2.2 Diffusion Models and Posterior Sampling

Diffusion models[1, 2, 3] are a class of generative models, which involve a forward diffusion process and a reverse generation process. The forward diffusion process gradually adds Gaussian noise to the data, and the reverse process is a learnable process that generates the data by sequentially denoising a noise sample. Follow the notation in the EDM framework [2], the probabilistic flow that describes the process from normal distribution with variance $\sigma(s_1)^2$ to target distribution $p(\mathbf{x}; \sigma(s_0))$ can be formulated as the following ordinary differential equation (ODE):

$$d\mathbf{x} = -\dot{\sigma}(s)\sigma(s)\nabla_{\mathbf{x}} \log p(\mathbf{x}; \sigma(s))ds, \tag{3}$$

where $\nabla_{\mathbf{x}} \log p(\mathbf{x}; \sigma(s))$ is known as the score function and can be estimated by denoiser $D_\theta(\mathbf{x}, \sigma(s))$, i.e., $\nabla_{\mathbf{x}} \log p(\mathbf{x}; \sigma(s)) \approx (D_\theta(\mathbf{x}; \sigma(s)) - \mathbf{x})/\sigma(s)^2$. $\sigma(s)$ is the schedule function that controls the noise level with respect to diffusion step $s$, and it is set to $\sigma(s) = s$ in the EDM framework.

In many real-world applications, $\mathbf{x}$ is not directly observable and we can only collect the measurements $\mathbf{y}$ with likelihood $p(\mathbf{y}|\mathbf{x}) = \mathcal{N}(\mathbf{y}; \mathcal{A}(\mathbf{x}), \varepsilon^2 \mathbf{I})$, where $\mathcal{A}(\cdot)$ is the measure operator and $\mathbf{I}$ is the identity matrix. We aim to infer or sample from the posterior distribution $p(\mathbf{x}|\mathbf{y})$, which is a common practice of inverse problem in various fields. The work[9] proposed diffusion posterior sampling (DPS), a widely-used method to add guidance gradient to the vanilla score function and enable conditional generation:

$$d\mathbf{x} = -\dot{\sigma}(s)\sigma(s)(\nabla_{\mathbf{x}} \log p(\mathbf{x}; \sigma(s)) + \nabla_{\mathbf{x}} \log p(\mathbf{y}|\mathbf{x}; \sigma(s)))ds. \tag{4}$$

Then the guidance gradient with respect to the log-likelihood function can be approximated with [9]:

$$\nabla_{\mathbf{x}_s} \log p(\mathbf{y}|\mathbf{x}_s; \sigma(s)) \approx \nabla_{\mathbf{x}_s} \log p(\mathbf{y}|D_\theta(\mathbf{x}_s); \sigma(s)) \approx -\frac{1}{\varepsilon^2} \nabla_{\mathbf{x}_s} \|\mathbf{y} - \mathcal{A}(D_\theta(\mathbf{x}_s))\|_2^2 . \quad (5)$$

Here, $D_\theta(\mathbf{x}_s)$ denotes the estimation of the denoised data at denoising step $s$.

## 3 Methodology

**Problem Statement:** We consider a $K$-mode physical full field tensor with an extra time mode $\mathcal{Y}$, whose spatiotemporal coordinate is denoted as $(\mathbf{i}, t) = (i_1, \cdots, i_K, t)$. Without loss of generality, suppose we have $M$ offgrid observation timesteps $\mathcal{T} = \{t_1, \ldots, t_M\}$. At each $t_m$, the observation set $\mathcal{O}_{t_m} = \{(\mathbf{i}_{n_m}, t_m, y_{\mathbf{i}_{n_m}, t_m})\}_{n_m=1}^{N_m}$ contains $N_m$ samples, where $y_{\mathbf{i}_{n_m}, t_m}$ is the entry value of $\mathcal{Y}$ indexed at $(\mathbf{i}_{n_m}, t_m)$. We denote the entire collection of observations by $\mathcal{O} = \{\mathcal{O}_{t_m}\}_{m=1}^M$.

We assume the patterns of observations at each time step is varying. The goal is to infer the value of underlying function $y(\mathbf{i}, t) : \mathbb{R}^K \times \mathbb{R}_+ \to \mathbb{R}$ at arbitrary continuous spatiotemporal coordinates. If we treat the problem as a function approximation task, it can be solved by numerical methods [13, 14] and functional tensor methods [11, 12].

However, in high-dimensional settings, directly approximating such a function from sparse observations is challenging due to the curse of dimensionality[15]. To address this, we adopt a two-stage framework commonly used in the generative physics simulation [4, 5, 16, 17]. In the first stage, we train a diffusion model using observations sampled from $B$ batches of homogeneous dynamics, denoted as $\mathcal{D} = \{\mathcal{O}^b\}_{b=1}^B$, where $\mathcal{O}^b$ is an instance of $\mathcal{O}$ as mentioned before. In the second stage, given observations of the target physical dynamics, we employ the pretrained diffusion model as a data-driven prior and perform posterior sampling to generate the full-field evolution.

Compared with existing works [4, 5, 18, 6, 9], our setting is unique in two key aspects: 1) the training data is both sparse and off-grid across the spatiotemporal domain; 2) the objective is to generate the full-field evolution over a continuous spatiotemporal domain rather than on a discrete mesh grid. These characteristics introduce two main challenges. First, during model training, a more general and flexible latent space mapping is required to effectively represent the sparse and irregular observations. Second, during posterior inference, it is crucial to leverage the limited-time-step observations as guidance to generate the continuous full-field evolution.

Therefore, we propose SDIFT, a novel sequential diffusion framework that integrates the functional Tucker Model, Gaussian process-based sequential diffusion, and message-passing diffusion posterior sampling. A schematic illustration of our approach is presented in Fig. 1.

### 3.1 Functional Tucker Model (FTM) as Universal Latent Representer

Without loss of generality, we consider a single batch of training data $\mathcal{O}^b = \{\mathcal{O}_{t_1}^b, \cdots, \mathcal{O}_{t_M}^b\}$. For notation simplicity, we omit the superscript $b$. Our goal is to map this sequence into a structured latent space. To achieve this, we adapt the Functional Tucker Model (FTM), a general framework that naturally captures the inherent multi-dimensional structure of physical fields and provides compact representations well-suited for sparse or irregular scenarios.

We treat $\mathcal{O}$ as irregular sparse observations of a $K$-mode functional tensor with an extra time mode $\mathcal{Y}$. Then, aligned with functional Tucker model (2), we assume the tensor sequence shares latent functions $\{\mathbf{f}_{\boldsymbol{\theta}_k}(i_k) : \mathbb{R} \to \mathbb{R}^{R_k}\}_{k=1}^K$ parameterized by $K$ neural networks, but with different learnable Tucker core $\{\mathcal{W}_{t_m} \in \mathbb{R}^{R_1 \times \cdots \times R_K}\}_{m=1}^M$. Specifically, for each observed entry $(\mathbf{i}, t_m, y_{\mathbf{i}, t_m}) \in \mathcal{O}_{t_m}$, it can be represented as:

$$y_{\mathbf{i}, t_m} \approx \text{vec}(\mathcal{W}_{t_m})^\top \left(\mathbf{f}_{\boldsymbol{\theta}_1}^1(i_1) \otimes \cdots \otimes \mathbf{f}_{\boldsymbol{\theta}_K}^K(i_K)\right). \quad (6)$$

After solving the above representation tasks for all entries in $\mathcal{O}$, we treat the estimated core sequence $\mathcal{W} := \{\mathcal{W}_{t_m}\}_{m=1}^M$ as the latent representation of $\mathcal{O}$. This process serves as an encoder, analogous to that in a variational autoencoder (VAE) [19], which maps the sparse observations to random variables in a latent functional Tucker space. On the other hand, given sampled core tensors $\mathcal{W}_{t_m}$ and the latent functions, Equation (6) functions as a decoder, allowing inference of $y(\mathbf{i}, t_m)$ at arbitrary spatial coordinate $\mathbf{i}$.

For the completeness of the paper, we further claim and prove that the *Universal Approximation Property (UAP)* of FTM. **Theorem 1.** *(UAP of FTM) Let $X_1, \cdots, X_K$ be compact subsets of $\mathbb{R}^K$.*

*Choose $u \in L^2(X_1 \times \cdots \times X_K)$. Then, for arbitrary $\epsilon > 0$, there exists sufficiently large $\{R_1 > 0, \cdots, R_K > 0\}$, coefficients $\{a_{r_1, \cdots, r_K}\}_{r_1, \cdots, r_K}^{R_1, \cdots, R_K}$ and neural networks $\{\{f_{r_k}^k\}_{r_k}^{R_k}\}_k^K$ such that*

$$\left\| u - \sum_{r_1}^{R_1} \cdots \sum_{r_K}^{R_K} [a_{r_1, \cdots, r_K} \prod_{k=1}^K f_{r_k}^k] \right\|_{L^2(X_1 \times \cdots \times X_K)} < \epsilon. \tag{7}$$

The detailed proof of **Theorem 1** is given in Appx. A. Theorem 1 states that the functional Tucker model can approximate any function in $L^2$ over the $K$-dimensional input space, thereby ensuring its expressive power for representing multidimensional physical fields. Moreover, consistent with the low-rank structure in physical fields[20, 21], the learned cores capture interactions among the $K$ latent functions, promoting computational efficiency [11] and offering a more interpretable representation [22] compared to nonlinear encoders like VAEs [19] or FiLM [23].

To train the latent functions $\{\mathbf{f}_{\boldsymbol{\theta}_k}\}_{k=1}^K$ and obtain batches of core sequences $\{\mathcal{W}^b\}_{b=1}^B$, we propose to minimize following objective function over the entire training dataset $\mathcal{D} = \{\mathcal{O}^b\}_{b=1}^B$:

$$\mathcal{L}_{\text{FTM}} = \mathbb{E}_{(\mathbf{i}, t_m, y_{\mathbf{i}, t_m}) \sim \mathcal{D}} \left[ \left\| y_{\mathbf{i}, t_m} - \text{vec}(\boldsymbol{\mathcal{W}}_{t_m})^\top \left( \mathbf{f}_{\boldsymbol{\theta}_1}^1(i_1) \otimes \cdots \otimes \mathbf{f}_{\boldsymbol{\theta}_K}^K(i_K) \right) \right\|_2^2 + \beta \text{TV}(\mathcal{W}) \right], \quad (8)$$

where $\text{TV}(\mathcal{W})^2$ denotes total variation regularization [24] over the core sequence on the temporal mode to ensure their temporal coherence, controlled by coefficient $\beta$. The training process is conducted using an alternating-direction strategy that iteratively updates latent functions and the core tensor sequence.

After training, we obtain a set of $K$ learned latent functions $\{\mathbf{f}_{\boldsymbol{\theta}_k}^k(\cdot)\}_{k=1}^K$ that summarize universal coordinate representations, and batches of learned core sequence $\{\mathcal{W}^b\}_{b=1}^B$, which capture the unique characteristics of the corresponding physical fields.

### 3.2 Gaussian Process based Sequential Diffusion for Core Sequence Generation

Given $\{\mathcal{W}^b\}_{b=1}^B$ from FTM training, we aim to build a diffusion model that generates core sequences with cores extracted at arbitrary timestep. For notation simplicity, we omit the superscript $b$. Assume that $\mathcal{W} = \{\boldsymbol{\mathcal{W}}_{t_1}, \cdots, \boldsymbol{\mathcal{W}}_{t_M}\}$ is drawn from an underlying continuous function $\boldsymbol{\mathcal{W}}(t) : \mathbb{R}_+ \to \mathbb{R}^{R_1 \times \cdots \times R_K}$ at $\mathcal{T} = \{t_1, \cdots, t_M\}$, where each element can be any continuous timestep. To model the distribution of $\boldsymbol{\mathcal{W}}(t)$, we follow the idea of functional diffusion model [25, 26], and choose a Gaussian Process (GP) noise [15] as the diffusion noise source. This approach naturally handles irregularly-sampled timesteps and incorporates temporal-correlated perturbations, i.e., assign varying noise level at different positions in the sequence, enabling the generation of temporal-continuous sequence samples.

Specifically, we assume the noise sequence $\mathcal{E} = \{\boldsymbol{\mathcal{E}}_{t_1}, \cdots, \boldsymbol{\mathcal{E}}_{t_M}\}$ is sampled from a GP $\mathcal{GP}(\mathbf{0}, \kappa(t_i, t_j))$, where $\kappa(t_i, t_j) = \exp\left(-\gamma(t_i - t_j)^2\right)$ is set as the radial basis function (RBF) kernel to model the temporal correlation of noise in sequence. As vanilla GP can only model the scaler-output function, here we actually build multiple independent GPs for each element of the core.

In the forward process of diffusion, a sample of core sequence $\mathcal{W}$ is perturbed by the GP noise $\mathcal{E}$. During the reverse process, the noise sequence from GP is then gradually denoised by a preconditioned neural network $D_\theta$ to obtain the clean core sequence. Notably, when $M = 1$, the model degenerates to the standard diffusion model (3). We refer to this model as Gaussian Process based Sequential Diffusion (GPSD).

Then, the training objective for the GPSD model is:

$$\mathcal{L}_{\text{GPSD}} = \mathbb{E}_{\sigma(s), \boldsymbol{\mathcal{W}}_{t_m} \sim \mathcal{W}, \boldsymbol{\mathcal{E}}_{t_m} \sim \mathcal{E}, t \sim \mathcal{T}} \left[ \lambda(\sigma(s)) \left\| D_\theta(\boldsymbol{\mathcal{W}}_{t_m} + \boldsymbol{\mathcal{E}}_{t_m}; \sigma(s), t_m) - \boldsymbol{\mathcal{W}}_{t_m} \right\|_2^2 \right], \quad (9)$$

where $\lambda(\sigma(s))$ is a weighting function dependent on the noise level $\sigma(s)$. Note that we aim to learn the distribution of a single core rather than the entire sequence of cores, allowing for flexible modeling of core sequence length and improved computational efficiency. To better capture temporal correlations within the sequence, we propose a temporally augmented U-Net (details in Appx. B) and summarize its training procedure in Appx. D.1.

---

$^2\text{TV}(\mathcal{W}) = \sum_{m=2}^M \left\| \boldsymbol{\mathcal{W}}_{t_m} - \boldsymbol{\mathcal{W}}_{t_{m-1}} \right\|_2^2.$

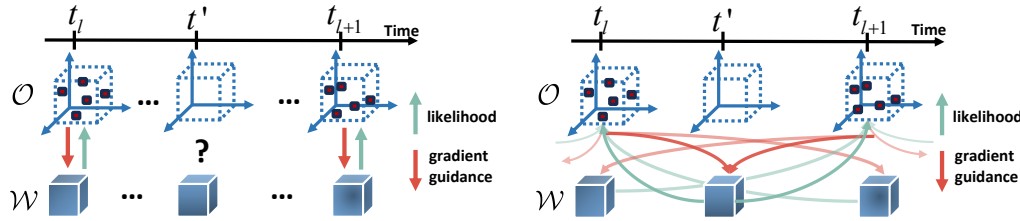

(a) DPS : Guidance missing at $t'$.      (b) MPDPS : Collected message as guidance at $t'$.

Figure 2: Illustration of DPS and MPDPS to handle the case of no observations at $t'$.

### 3.3 Message-Passing Diffusion Posterior Sampling

With pre-trained GPSD and FTM, we generate the core tensor sequence from GP noise and reconstruct dynamics at arbitrary $(\mathbf{i}, t)$ via FTM. We further introduce a posterior sampling method for conditional sequence generation from limited observations.

Consider the task of inferring dynamics at a target time set $\mathcal{T}_{\text{tar}} = \{t_1, \cdots, t_M\}$. With a pre-trained FTM, this reduces to generating the core tensor sequence $\mathcal{W} = \{\boldsymbol{\mathcal{W}}_{t_1}, \cdots, \boldsymbol{\mathcal{W}}_{t_M}\}$ from the observation set $\mathcal{O}$. We assume that observations are only available at a subset $\mathcal{T}_{\text{obs}} \subseteq \mathcal{T}_{\text{tar}}$, with $|\mathcal{T}_{\text{obs}}| = L \leq M$. Under standard DPS, only the observed cores $\mathcal{W}_{\text{obs}} := \{\boldsymbol{\mathcal{W}}_{t_l} \mid t_l \in \mathcal{T}_{\text{obs}}\}$ are guided by their corresponding observations $\mathcal{O}_{t_l}$, while the remaining cores are generated without guidance, as illustrated in Fig. 2(a). This is suboptimal, as real-world physical dynamics are temporally continuous and correlated, meaning that even limited observations can provide valuable guidance for the entire sequence.

To enable posterior sampling of the entire core sequence, we propose *Message-Passing Diffusion Posterior Sampling* (MPDPS). T*he key idea is to exploit the temporal continuity of the core sequence to smooth and propagate guidance from limited observations across the full sequence, as illustrated in Fig. 2(b).*

Without loss of generality, we focus on the timestep with observation $t_l \in \mathcal{T}_{\text{obs}}$ as a representative case, and demonstrate the propagation mechanism of how guidance message of $\mathcal{O}_{t_l}$ pass to the residual core sequence $\mathcal{W}_{\backslash t_l} = \mathcal{W} \setminus \boldsymbol{\mathcal{W}}_{t_l}$, where $|\mathcal{W}_{\backslash t_l}| = M - 1$. At diffusion step $s$, the guidance message group of $\mathcal{O}_{t_l}$ passed to $\mathcal{W}_{\backslash t_l}$ can be expressed as:

$$\{\boldsymbol{\mathcal{G}}^s_{t_l, t_m}\}_{t_m \in \mathcal{T}_{\text{tar}} \backslash t_l} := \nabla_{\mathcal{W}^s_{\backslash t_l}} \log p(\mathcal{O}_{t_l} | \mathcal{W}^s_{\backslash t_l}) = \nabla_{\mathcal{W}^s_{\backslash t_l}} \log \int p(\mathcal{O}_{t_l} | \boldsymbol{\mathcal{W}}^0_{t_l}) p(\boldsymbol{\mathcal{W}}^0_{t_l} | \mathcal{W}^s_{\backslash t_l}) \mathrm{d}\boldsymbol{\mathcal{W}}^0_{t_l},$$

(10)

where $\boldsymbol{\mathcal{G}}^s_{t_l, t_m} = \nabla_{\boldsymbol{\mathcal{W}}^s_{t_m}} \log p(\mathcal{O}_{t_l} | \mathcal{W}^s_{\backslash t_l})$ is the likelihood gradient of $\mathcal{O}_{t_l}$ respect to $\boldsymbol{\mathcal{W}}^s_{t_m}$, $\mathcal{W}^s_{\backslash t_l}$ denotes perturbed $\mathcal{W}_{\backslash t_l}$ at diffusion step $s$, $\nabla_{\mathcal{W}^s_{\backslash t_l}}$ is the operator to compute the gradient respect to each core in $\mathcal{W}^s_{\backslash t_l}$, and $\boldsymbol{\mathcal{W}}^0_{t_l}$ denotes the clean core at timestep $t_l$. However, directly computing (10) is intractable. While $p(\mathcal{O}_{t_l} | \boldsymbol{\mathcal{W}}^0_{t_l})$ is a Gaussian factor due to the multilinear structure of the FTM, the conditional distribution $p(\boldsymbol{\mathcal{W}}^0_{t_l} | \mathcal{W}^s_{\backslash t_l})$ is unknown, making the integration term intractable.

To handle this, we further assign element-wise GP prior over the core sequence, consistent with generation by GP noise in Sec. 3.2, and claim that $p(\boldsymbol{\mathcal{W}}^0_{t_l} | \mathcal{W}^s_{\backslash t_l})$ can be estimated as a Gaussian factor via the denoiser network $D_\theta$ and Gaussian process regression [15] (GPR). Then we can work out an tractable approximation of (10) as follows:

$$\nabla_{\mathcal{W}^s_{\backslash t_l}} \log p(\mathcal{O}_{t_l} | \mathcal{W}^s_{\backslash t_l}) \approx \nabla_{\mathcal{W}^s_{\backslash t_l}} [-\frac{1}{2} (\mathbf{y}_{t_l} - \mathbf{B}_{t_l} T(\mathcal{W}^s_{\backslash t_l}))^\top \tilde{\boldsymbol{\Sigma}}^{-1}_{t_l} (\mathbf{y}_{t_l} - \mathbf{B}_{t_l} T(\mathcal{W}^s_{\backslash t_l}))], \quad (11)$$

where $\mathbf{y}_{t_l}$ is vector collecting entries in $\mathcal{O}_{t_l}$, and $T(\mathcal{W}^s_{\backslash t_l})$ with shape of $(M-1) \times (\prod_{k=1}^K R_k)$ is the concatenation of vectorized estimated clean $\boldsymbol{\mathcal{W}}^0_{t_m} \in \mathcal{W}_{\backslash t_l}$ from $D_\theta$, which takes every $\boldsymbol{\mathcal{W}}^s_{t_m} \in \mathcal{W}^s_{\backslash t_l}$ as input. $\mathbf{B}_{t_l}$ and $\tilde{\boldsymbol{\Sigma}}_{t_l}$ are constructed by the latent functions in FTM and the GPR kernel matrix in a closed form. A full derivation is provided in Appx. C. Note that (11) has a quadratic form, enabling efficient gradient computation.

At diffusion step $s$, we go through all observations, compute the guidance messages to be passed with (11), and for core $\boldsymbol{\mathcal{W}}_{t_n}^s$ at $t_n \in \mathcal{T}_{\text{tar}}$, the guidance messages from observations are:

$$\nabla_{\boldsymbol{\mathcal{W}}_{t_n}^s} \log p(\mathcal{O}|\boldsymbol{\mathcal{W}}_{t_n}^s) \approx \mathbf{1}_{\mathcal{T}_{\text{obs}}}(t_n) \cdot \nabla_{\boldsymbol{\mathcal{W}}_{t_n}^s} \log p(\mathcal{O}_{t_n}|\boldsymbol{\mathcal{W}}_{t_n}^s) + \sum_{t_l \in \mathcal{T}_{\text{obs}} \setminus t_n} \boldsymbol{\mathcal{G}}_{t_l, t_n}^s, \quad (12)$$

where $\mathbf{1}_{\mathcal{T}_{\text{obs}}}(t_n) = 1$ if $t_n \in \mathcal{T}_{\text{obs}}$, otherwise 0, and $\nabla_{\boldsymbol{\mathcal{W}}_{t_n}^s} \log p(\mathcal{O}_{t_n}|\boldsymbol{\mathcal{W}}_{t_n}^s)$ denotes the guidance for core at timestep with direct observations, which is equal to the DPS guidance in (5). Then we can plug (12) into (4) to perform posterior sampling.

As shown in (12), MPDPS propagates observation-derived guidance across the entire core sequence. For cores at timesteps with direct observations, this guidance is further smoothed by messages from other observed timesteps. This smoothing mechanism enhances the robustness of the generated sequence, especially under noisy or extremely sparse observations. Consequently, MPDPS can outperform standard DPS even when $\mathcal{T}_{\text{tar}} = \mathcal{T}_{\text{obs}}$, as demonstrated in our experiments. We provide a more detailed explanation of the MPDPS mechanism in Appx. E to facilitate understanding. The full MPDPS procedure is summarized in Algo. 1.

---

**Algorithm 1** Message-passing Diffusion Posterior Sampling

**Require:** Deterministic Sampler $D_\theta(\cdot; \sigma(s))$, noise power schedule $\{\sigma(s_i)\}_{i=0}^S$, observation set $\mathcal{O}$, target time set $\mathcal{T}_{\text{tar}}$, observation time set $\mathcal{T}_{\text{obs}}$, kernel parameter $\gamma$. Weights $\zeta$.
1: Generate initial GP noise sequence $\{\boldsymbol{\mathcal{W}}_{t_1}^S, \cdots, \boldsymbol{\mathcal{W}}_{t_M}^S\}$ at $\mathcal{T}_{\text{tar}}$ with kernel function $\mathbf{K}_{i,j} = \exp\left(-\gamma(t_i - t_j)^2\right)$.
2: **for** $i = S, \ldots, 1$ **do**
3:     Estimate all denoised cores at step $s_i$: $\hat{\boldsymbol{\mathcal{W}}}_{t_j}^0 \leftarrow D_\theta\left(\boldsymbol{\mathcal{W}}_{t_j}^i; \sigma(s_i)\right), \forall j$.
4:     Evaluate $d\boldsymbol{\mathcal{W}}/d\sigma(s)$ at diffusion step $s_i$: $\boldsymbol{\mathcal{D}}_j^i \leftarrow (\boldsymbol{\mathcal{W}}_{t_j}^i - \hat{\boldsymbol{\mathcal{W}}}_{t_j}^0)/\sigma(s_i), \forall j$.
5:     Euler step: $\boldsymbol{\mathcal{W}}_{t_j}^{i-1} \leftarrow \boldsymbol{\mathcal{W}}_{t_j}^i + \left(\sigma(s_{i-1}) - \sigma(s_i)\right) \boldsymbol{\mathcal{D}}_j^i, \forall j$.
6:     **if** $\sigma(t_{i-1}) \neq 0$ **then**
7:         Secondorder correction: $\hat{\boldsymbol{\mathcal{W}}}_{t_j}^0 \leftarrow D_\theta\left(\boldsymbol{\mathcal{W}}_{t_j}^{i-1}; \sigma(s_{i-1})\right), \forall j$.
8:         Evaluate $d\boldsymbol{\mathcal{W}}/d\sigma(s)$ at diffusion step $s_{i-1}$: $\boldsymbol{\mathcal{D}}_j^{'i} \leftarrow (\boldsymbol{\mathcal{W}}_{t_j}^{i-1} - \hat{\boldsymbol{\mathcal{W}}}_{t_j}^0)/\sigma(s_{i-1}), \forall j$.
9:         Apply the trapezoidal rule at diffusion step $s_{i-1}$: $\boldsymbol{\mathcal{W}}_{t_j}^{i-1} \leftarrow \boldsymbol{\mathcal{W}}_{t_j}^i + \left(\sigma(t_{i+1}) - \sigma(t_i)\right)\left(\frac{1}{2}\boldsymbol{\mathcal{D}}_j^i + \frac{1}{2}\boldsymbol{\mathcal{D}}_j^{'i}\right), \forall j$
10:     **end if**
11:     Compute the guidance gradients as defined in (12) for all target cores, and accumulate each into its corresponding core.
12: **end for**
13: **return** Guided generated core sequence $\{\boldsymbol{\mathcal{W}}_{t_1,0}, \cdots, \boldsymbol{\mathcal{W}}_{t_M,0}\}$.

---

## 4 Related Work

**Tensor-based methods** are widely used to reconstruct multidimensional data by learning low-rank latent factors from sparse observations. Recent works in temporal tensor learning [27, 28, 22, 29] relax the discretization constraints of classical tensors, enabling modeling of continuous temporal dynamics. Functional tensor approaches [12, 11, 30, 31] go further by modeling continuity across all tensor modes via function approximation with structured latent factors. While effective, these methods are limited to single-instance training and inference, and lack the flexibility to handle sets of tensors with varying dynamics, e.g., PDE trajectories under different initial conditions.

Toward generative physical simulation, several works [9, 17, 16, 4, 6] have applied pre-trained **diffusion models**, achieving notable progress across various tasks. However, most existing methods assume well-structured data and struggle with real-world reconstruction scenarios involving sparse and irregular observations. To address this, [5] proposed modeling off-grid data using conditional neural fields (CNFs) followed by diffusion. Yet, this approach do not explicitly capture temporal continuity, limiting their ability to generate full-field dynamics. Additionally, it suffers from high computational cost due to repeated gradient backpropagation operated on CNF during posterior

| Methods | Supernova Explosion | | Ocean Sound Speed | | Active Matter | |
|---|---|---|---|---|---|---|
| | $\rho = 1\%$ | $\rho = 3\%$ | $\rho = 1\%$ | $\rho = 3\%$ | $\rho = 1\%$ | $\rho = 3\%$ |
| **Observation setting 1** : $\mathcal{T}_{obs} = \mathcal{T}_{tar}$ | | | | | | |
| *Tensor-based* | | | | | | |
| LRTFR [11] | $0.558 \pm 0.044$ | $0.429 \pm 0.043$ | $0.345 \pm 0.036$ | $0.217 \pm 0.066$ | $0.302 \pm 0.104$ | $0.258 \pm 0.022$ |
| DEMOTE [38] | $1.285 \pm 0.102$ | $1.213 \pm 0.217$ | $0.358 \pm 0.127$ | $0.314 \pm 0.086$ | $0.950 \pm 0.486$ | $0.871 \pm 0.497$ |
| NONFAT [27] | $1.229 \pm 0.127$ | $1.197 \pm 0.204$ | $0.402 \pm 0.090$ | $0.330 \pm 0.101$ | $0.921 \pm 0.457$ | $0.867 \pm 0.413$ |
| *Attention-based* | | | | | | |
| Senseiver [18] | $0.446 \pm 0.041$ | $0.349 \pm 0.023$ | $0.264 \pm 0.037$ | $0.2005 \pm 0.031$ | $0.345 \pm 0.094$ | $0.264 \pm 0.076$ |
| *Diffusion-based* | | | | | | |
| CoNFiLD [5] | $0.561 \pm 0.082$ | $0.427 \pm 0.037$ | $0.201 \pm 0.034$ | $0.145 \pm 0.012$ | $0.529 \pm 0.087$ | $0.5075 \pm 0.830$ |
| SDIFT w/ DPS | $0.339 \pm 0.116$ | $0.291 \pm 0.033$ | $0.194 \pm 0.073$ | $0.160 \pm 0.035$ | $0.298 \pm 0.065$ | $0.174 \pm 0.043$ |
| SDIFT w/ MPDPS | $\mathbf{0.283 \pm 0.026}$ | $\mathbf{0.272 \pm 0.025}$ | $\mathbf{0.146 \pm 0.046}$ | $\mathbf{0.108 \pm 0.043}$ | $\mathbf{0.215 \pm 0.068}$ | $\mathbf{0.156 \pm 0.046}$ |
| **Observation setting 2** : $\|\mathcal{T}_{obs}\| = \frac{1}{2}\|\mathcal{T}_{tar}\|$ | | | | | | |
| *Tensor-based* | | | | | | |
| LRTFR [11] | $0.783 \pm 0.416$ | $0.813 \pm 0.296$ | $0.610 \pm 0.323$ | $0.508 \pm 0.297$ | $0.620 \pm 0.484$ | $0.598 \pm 0.527$ |
| DEMOTE [38] | $1.351 \pm 0.209$ | $1.223 \pm 0.397$ | $0.549 \pm 0.181$ | $0.533 \pm 0.198$ | $1.261 \pm 0.614$ | $1.277 \pm 0.603$ |
| NONFAT [27] | $1.278 \pm 0.214$ | $1.254 \pm 0.2785$ | $0.465 \pm 0.153$ | $0.420 \pm 0.189$ | $1.126 \pm 0.514$ | $1.270 \pm 0.485$ |
| *Attention-based* | | | | | | |
| Senseiver [18] | Not capable | - | - | - | - | - |
| *Diffusion-based* | | | | | | |
| CoNFiLD [5] | $0.757 \pm 0.199$ | $0.6575 \pm 0.148$ | $0.310 \pm 0.054$ | $0.2615 \pm 0.038$ | $0.8265 \pm 0.167$ | $0.779 \pm 0.161$ |
| SDIFT w/ DPS | $0.659 \pm 0.057$ | $0.6450 \pm 0.054$ | $0.412 \pm 0.156$ | $0.407 \pm 0.136$ | $0.674 \pm 0.153$ | $0.637 \pm 0.113$ |
| SDIFT w/ MPDPS | $\mathbf{0.433 \pm 0.163}$ | $\mathbf{0.335 \pm 0.122}$ | $\mathbf{0.181 \pm 0.084}$ | $\mathbf{0.165 \pm 0.041}$ | $\mathbf{0.296 \pm 0.096}$ | $\mathbf{0.256 \pm 0.087}$ |

Table 1: VRMSEs of the reconstruction results for all methods across three datasets, evaluated under two observation settings with different observation ratios.

sampling. [18] introduced an attention-based model for reconstructing fields from sparse, off-grid measurements, but it lacks explicit temporal modeling and struggles with rapidly evolving fields.

In parallel, **operator learning** methods [32, 33] have emerged as powerful tools for physical modeling by learning mappings between functions, and have been applied to field reconstruction tasks [34, 35, 36, 37]. However, these methods often rely on task-specific designs and predefined observation patterns, limiting their adaptability. Notably, [32] reported that FNOs can perform poorly in high-dimensional settings with sparse data.

To address these gaps, we propose SIDFT, a method well-suited for high-dimensional, continuous spatiotemporal field reconstruction in sparse and irregular settings, with strong flexibility and performance across domains and scales.

## 5 Experiment

**Datasets:** We examined SDIFT on three real-world benchmark datasets, which span across astronomical, environmental and molecular scales. (1) *Supernova Explosion*, temperature evolution of a supernova blast wave in a compressed dense cool monatomic ideal-gas cloud. We extracted 396 records in total, each containing 16 frames, where each frame has a shape of $64 \times 64 \times 64$. We use 370 for training, randomly masking out $85\%$ of the points in each record to simulate irregular sparse data. The remaining 26 records are reserved for testing.(https://polymathic-ai.org/the_well/datasets/supernova_explosion_64/; (2) *Ocean Sound Speed*, sound speed field measurements in the pacific ocean. We extracted 1000 records of shape $24 \times 5 \times 38 \times 76$, using 950 for training (with 90% of points randomly masked) and reserving 50 for testing. (https://ncss.hycom.org/thredds/ncss/grid/GLBy0.08/expt_93.0/ts3z/dataset.html). (3) *Active Matter*, the dynamics of rod-like active particles in a stokes fluid simulated via a continuum theory. We extracted 928 records, each of size $24 \times 256 \times 256$, using 900 for training (with 90% of points randomly masked) and reserving 28 for testing.(https://polymathic-ai.org/the_well/datasets/active_matter/)

**Baselines and Settings:** We compared SDIFT with state-of-the-art physical field reconstruction methods and tensor-based methods: (1) Senseiver [18], an attention-based framework that encodes sparse sensor measurements into a unified latent space for efficient multidimensional field reconstruction; (2) CoNFiLD [5], a generative model that combines conditional neural field encoding with latent diffusion to generate high-fidelity fields. (3) LRTFR [11], a low-rank functional Tucker model employing factorized neural representations for tensor decomposition; (4) DEMOTE [38], a neural diffusionreaction process model that learns dynamic factors within a tensor decomposition

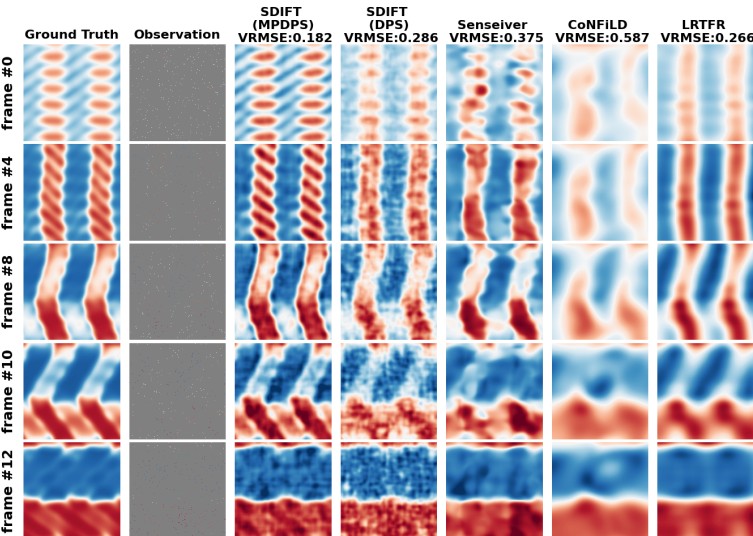

Figure 3: Reconstruction of *Active Matter* dynamics under **observation setting 1** with $\rho = 1\%$. SDIFT guided by MPDPS produces notably robust reconstructions with fine-grained details.

framework; (5) NONFAT [27], a bi-level latent Gaussian process model that estimates time-varying factors using Fourier bases; We evaluate our method against baselines on reconstructing temporal physical fields under two observation settings with observation ratio $\rho \in \{1\%, 3\%\}$. **In observation setting 1**, $\mathcal{T}_{\mathrm{obs}} = \mathcal{T}_{\mathrm{tar}}$, so at each $t_i$ observations occur at ratio $\rho$. **In observation setting 2**, $\mathcal{T}_{\mathrm{obs}} \subset \mathcal{T}_{\mathrm{tar}}$: for $t_i \in \mathcal{T}_{\mathrm{obs}}$ we retain observations at ratio $\rho$, whereas for $t_i \in \mathcal{T}_{\mathrm{tar}} \setminus \mathcal{T}_{\mathrm{obs}}$ no observations are available. We assume that $\mathcal{T}_{\mathrm{obs}}$ is constructed by selecting every other element from $\mathcal{T}_{\mathrm{tar}}$ so that $|\mathcal{T}_{\mathrm{obs}}| = \frac{1}{2}|\mathcal{T}_{\mathrm{tar}}|$. We replace MPDPS with traditional DPS [9] to constitute an ablation study to showcase the effectiveness of our proposed MPDPS. The performance metrics is the Variance-scaled Root Mean Squared Error (VRMSE, see Appx. F.1 for definition), which offers a scale-independent way to evaluate model performance. Each experiment was conducted 10 times and we reported the average test errors with their standard deviations. We provided more implementation details in Appx. F.

**Main Evaluation Results:** The quantitative results in Tab. 1 demonstrate that SDIFT consistently outperforms all baselines by a substantial margin under both observation settings 1 and 2. We further observe that tensor-based methods generally lag behind training-based approaches, since they cannot exploit batches of historical data and must reconstruct the field from extremely sparse observations. In particular, temporal tensor techniques that ignore the continuously indexed modesuch as NON-FAT and DEMOTEperform poorly across all three datasets compared to LRTFR. The qualitative comparisons in Fig. 3 indicate that SDIFT with MPDPS produces more faithful reconstructions.

Observation setting 2 presents a more challenging scenario, in which all baselines exhibit significant performance degradation. CoNFiLD and SIDFT with DPS can still reconstruct via unconditional generation without any observations, whereas Senseiver cannot. As shown in Fig. 4, SDIFT guided by DPS generates physical fields randomly at timesteps without observations, owing to the absence of guidance in such cases. Similarly, Senseiver fails when observations are unavailable. In contrast, MPDPS effectively overcomes these limitations and consistently produces accurate reconstructions, highlighting the robustness of the proposed method. Additional unconditional generation results, conditional reconstruction results and detailed analysis on CoNFiLD are provided in Appx. G.1G.2G.3, respectively. We also demonstrated the effectiveness of using GP noise as the diffusion source in Appx. G.5. These results collectively demonstrate the effectiveness of SDIFT with the MPDPS mechanism.

**Sampling Speed:** We compared the sampling speed of our method with CoNFiLD [5] on a NVIDIA RTX 4090 GPU with 24 GB memory; the results are shown in Tab. 2. One can see that our proposed method is significantly faster than CoNFiLD across all settings. This rapid inference speed stems from two key factors. First, we formulated the reverse diffusion process as solving a deterministic probability-flow ODE rather than a stochastic differential equation (SDE, as in CoNFiLD), which largely reduces the number of neural network evaluations[2] during sampling. Second, by employ-

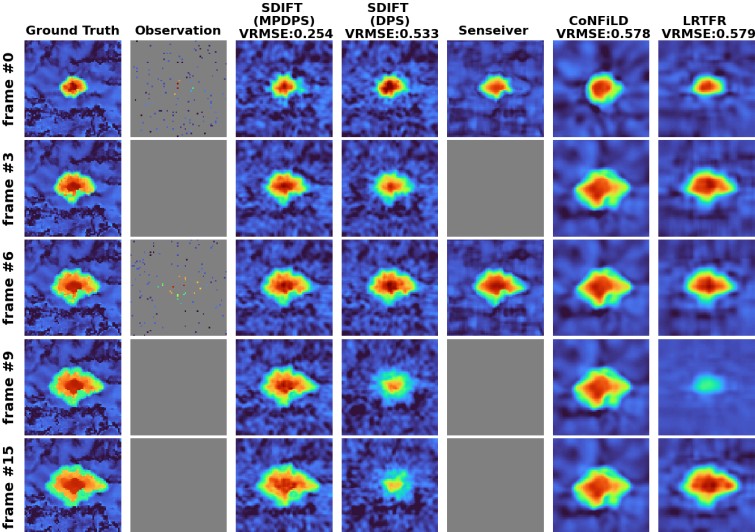

Figure 4: Reconstruction of *Supernova Explosion* dynamics under **observation setting 2** with $\rho = 3\%$. Since DPS does not provide guidance at timesteps without observations (**i.e., frames # 3,9,15**), SDIFT generates the corresponding physical fields randomly. In contrast, MPDPS effectively addresses this limitation and yields smooth reconstructions.

| Methods | Supernova Explosion | | | Ocean Sound Speed | | | Active Matter | | |
|---|---|---|---|---|---|---|---|---|---|
| | $\rho = 1\%$ | $\rho = 3\%$ | #Para. | $\rho = 1\%$ | $\rho = 3\%$ | #Para. | $\rho = 1\%$ | $\rho = 3\%$ | #Para. |
| CoNFiLD | 31.3s | 44.0s | 23M | 15.9s | 20.3s | 10M | 27.6s | 31.5s | 10M |
| SDIFT w/ MPDPS | **2.23s** | **5.43s** | 26M | **0.84s** | **0.89s** | 15M | **1.31s** | **1.42s** | 12M |

Table 2: Average sampling speed for reconstruction with different observation ratios on observation setting 1.

ing a functional Tucker model, we endowed the likelihood with a quadratic structure that enables efficient computation of posterior gradients. In contrast, CoNFiLD defines its likelihood through a conditional neural field and must compute gradients via automatic differentiation at each diffusion step, significantly increasing the computation time.

**Robustness against Noise:** We evaluated SDIFT with DPS and MPDPS against three noise types: Gaussian, Poisson and Laplacian, with different variance levels, and results are at Tab. 4 in Appx. G.4, which demonstrates the robustness introduced by proposed MSDPS module.

## 6  Conclusion and Future Work

We presented SDIFT, a generative model that efficiently captures the distribution of multidimensional physical dynamics from irregular and limited training data. We also proposed a novel message-passing diffusion posterior sampling mechanism for conditional generation with observations, achieving state-of-the-art reconstruction performance with significant computational efficiency. A current limitation of our work is the lack of explicit incorporation of physical laws into the modeling process. In future work, we plan to integrate SDIFT with domain-specific physical knowledge, enabling more accurate long-range and wide-area physical field reconstructions.

## 7  Acknowledgement

This work was supported in part by the National Natural Science Foundation of China under Grant 62371418 and Grant 62322606, in part by the Fundamental Research Funds for the Central Universities (226-2025-00168) and in part by Zhejiang University Education Foundation Qizhen Scholar Foundation.

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

# APPEXNDIX

## A   Proof of Theorem 1

Here we show the preliminary lemmas and proofs for Theorem 1. We consider the general tensor product between $K$ Hilbert spaces.

**Definition 1.** $\{v_{\alpha_i}\}$ is an orthonormal basis for Hilbert space $\mathcal{H}_i, \forall i$.

**Lemma 1.** $\{v_{\alpha_1} \otimes v_{\alpha_2}\}$ is an orthonormal basis for $\mathcal{H}_1 \otimes \mathcal{H}_2$.

*Proof.* See **Lemma 2.** in the Appendix A of [39]. $\qquad\square$

**Lemma 2.** $\{v_{\alpha_1} \otimes \cdots \otimes v_{\alpha_K}\}$ is an orthonormal basis for $\mathcal{H}_1 \otimes \cdots \otimes \mathcal{H}_K$.

*Proof.* By repeatedly applying **Lemma 2.**, we can show that $\{v_{\alpha_1} \otimes \cdots \otimes v_{\alpha_K}\}$ is an orthonormal basis for $\mathcal{H}_1 \otimes \cdots \otimes \mathcal{H}_K$. $\qquad\square$

**Theorem 1.** *(Universal Approximation Property) Let $X_1, \cdots, X_K$ be compact subsets of $\mathbb{R}^K$. Choose $u \in L^2(X_1 \times \cdots \times X_K)$. Then, for arbitrary $\epsilon > 0$, we can find sufficiently large $\{R_1 > 0, \cdots, R_K > 0\}$, coefficients $\{a_{r_1,\cdots,r_K}\}_{r_1,\cdots,r_K}^{R_1,\cdots,R_K}$ and neural networks $\{f_{r_1}^1, \cdots, f_{r_K}^K\}$ such that*

$$\left\| u - \sum_{r_1}^{R_1} \cdots \sum_{r_K}^{R_K} [a_{r_1,\cdots,r_K} \prod_{k=1}^{K} f_{r_k}^k] \right\|_{L^2(X_1 \times \cdots \times X_K)} < \epsilon. \tag{13}$$

*Proof.* Let $\{\phi_{r_1}\}, \cdots, \{\phi_{r_K}\}$ be an orthonormal basis for $L^2(X_1), \cdots, L^2(X_K)$, respectively. According to **Lemma 2**, $\{\phi_{r_1} \cdots \phi_{r_K}\}$ forms an orthonormal basis for $L^2(X_1 \times \cdots \times X_K)$. Therefore, we can find sufficiently large set $\{R_1 > 0, \cdots, R_K > 0\}$ and arbitrary $\epsilon > 0$ such that

$$\left\| u - \sum_{r_1}^{R_1} \cdots \sum_{r_K}^{R_K} [c_{r_1,\cdots,r_K} \prod_{k=1}^{K} \phi_{r_k}^k] \right\|_{L^2(X_1 \times \cdots \times X_K)} < \frac{\epsilon}{2}. \tag{14}$$

$\qquad\square$

Here, $c_{r_1,\cdots,c_K}$ is defined as

$$c_{r_1,\cdots,c_K} = \int u(i_1,\cdots,i_K) \prod_{k=1}^{K} \phi_{r_k}^k(i_k) \prod_{k=1}^{K} di_k. \tag{15}$$

Also, with the universal approximation theorem in [40], we can find neural networks $\{f_{r_k}^k\}_{k=1}^K$ satisfy

$$\left\| \phi_{r_k}^k - f_{r_k}^k \right\|_{L^2(X_k)} \le \frac{\epsilon}{(1 + 4\prod_{k=2}^{K} R_k)^{\frac{r_k}{2}} K \|u\|_{L^2(X_1 \times \cdots \times X_K)}}, \forall k. \tag{16}$$

First, we have

$$\left\| u - \sum_{r_1}^{R_1} \cdots \sum_{r_K}^{R_K} [c_{r_1,\cdots,r_K} \prod_{k=1}^{K} f_{r_k}^k] \right\|_{L^2(X_1 \times \cdots \times X_K)}$$

$$= \left\| u - \sum_{r_1}^{R_1} \cdots \sum_{r_K}^{R_K} [c_{r_1,\cdots,r_K} \prod_{k=1}^{K} \phi_{r_k}^k] + \sum_{r_1}^{R_1} \cdots \sum_{r_K}^{R_K} [c_{r_1,\cdots,r_K} \prod_{k=1}^{K} \phi_{r_k}^k] - \sum_{r_1}^{R_1} \cdots \sum_{r_K}^{R_K} [c_{r_1,\cdots,r_K} \prod_{k=1}^{K} f_{r_k}^k] \right\|_{L^2(X_1 \times \cdots \times X_K)}$$

$$\leq \underbrace{\left\| u - \sum_{r_1}^{R_1} \cdots \sum_{r_K}^{R_K} [c_{r_1,\cdots,r_K} \prod_{k=1}^{K} \phi_{r_k}^k] \right\|_{L^2(X_1 \times \cdots \times X_K)}}_{A_1} + \tag{17}$$

$$\underbrace{\left\| \sum_{r_1}^{R_1} \cdots \sum_{r_K}^{R_K} [c_{r_1,\cdots,r_K} \prod_{k=1}^{K} \phi_{r_k}^k] - \sum_{r_1}^{R_1} \cdots \sum_{r_K}^{R_K} [c_{r_1,\cdots,r_K} \prod_{k=1}^{K} f_{r_k}^k] \right\|_{L^2(X_1 \times \cdots \times X_K)}}_{A_2}.$$

The first term $A_1$ can be determined by Eq.(14). We are to estimate the second term.

$$\sum_{r_1}^{R_1} \cdots \sum_{r_K}^{R_K} [c_{r_1,\cdots,r_K} \prod_{k=1}^{K} \phi_{r_k}^k] - \sum_{r_1}^{R_1} \cdots \sum_{r_K}^{R_K} [c_{r_1,\cdots,r_K} \prod_{k=1}^{K} f_{r_k}^k]$$

$$= \underbrace{\sum_{r_1}^{R_1} \cdots \sum_{r_K}^{R_K} c_{r_1,\cdots,r_K} (\prod_{k=2}^{K} \phi_{r_k}^k)(\phi_{r_1}^1 - f_{r_1}^1)}_{B_1} + \underbrace{\sum_{r_1}^{R_1} \cdots \sum_{r_K}^{R_K} c_{r_1,\cdots,r_K} (f_{r_1}^1 \prod_{k=3}^{K} \phi_{r_k}^k)(\phi_{r_2}^2 - f_{r_2}^2)}_{B_2} + \cdots + \tag{18}$$

$$\underbrace{\sum_{r_1}^{R_1} \cdots \sum_{r_K}^{R_K} c_{r_1,\cdots,r_K} (\prod_{k=1}^{K-1} f_{r_k}^k)(\phi_{r_K}^K - f_{r_K}^K)}_{B_K}.$$

Then, we have

$$\begin{aligned} A_2 &= \|B_1 + \cdots + B_k\|_{L^2(X_1 \times \cdots \times X_K)} \\ &\leq \|B_1\|_{L^2(X_1 \times \cdots \times X_K)} + \cdots + \|B_K\|_{L^2(X_1 \times \cdots \times X_K)} \end{aligned} \tag{19}$$

Specifically, we have

$$\|B_1\|_{L^2(X_1 \times \cdots \times X_K)}^2 = \int \left\{ \sum_{r_1}^{R_1} \cdots \sum_{r_K}^{R_K} c_{r_1,\cdots,r_K} (\prod_{k=2}^{K} \phi_{r_k}^k)(\phi_{r_1}^1 - f_{r_1}^1) \right\}^2 \prod_{k=1}^{K} di_k$$

$$= \int \left\{ \sum_{r_1}^{R_1} \underbrace{\left[ \sum_{r_2}^{R_2} \cdots \sum_{r_K}^{R_K} c_{r_1,\cdots,r_K} (\prod_{k=2}^{K} \phi_{r_k}^k) \right]}_{C_1} \underbrace{(\phi_{r_1}^1 - f_{r_1}^1)}_{C_2} \right\}^2 \prod_{k=1}^{K} di_k. \tag{20}$$

By applying Cauchy-Scharwz inequality, we have

$$\|B_1\|_{L^2(X_1 \times \cdots \times X_K)}^2 \leq \int \left( \sum_{r_1}^{R_1} |C_1|^2 \right) \left( \sum_{r_1}^{R_1} |C_2|^2 \right) \prod_{k=1}^{K} di_k$$

$$= \underbrace{\left( \int \sum_{r_1}^{R_1} |C_1|^2 \prod_{k=2}^{K} di_k \right)}_{D_1} \underbrace{\left( \int \sum_{r_1}^{R_1} |C_2|^2 di_1 \right)}_{D_2}. \tag{21}$$

Thereafter, we have

$$D_1 = \int \sum_{r_1}^{R_1} |C_1|^2 \prod_{k=2}^{K} di_k = \sum_{r_1}^{R_1} \int |C_1|^2 \prod_{k=2}^{K} di_k = \sum_{r_1}^{R_1} \left\| \sum_{r_2}^{R_2} \cdots \sum_{r_K}^{R_K} c_{r_1, \cdots, r_K} (\prod_{k=2}^{K} \phi_{r_k}^k) \right\|_{L^2(X_2 \times \cdots X_K)}^2 \tag{22}$$

Since $\{\phi_{r_k}^k\}_{k=1}^{K}$ are all orthonormal basis, we have

$$\left\| \sum_{r_2}^{R_2} \cdots \sum_{r_K}^{R_K} c_{r_1, \cdots, r_K} (\prod_{k=2}^{K} \phi_{r_k}^k) \right\|_{L^2(X_2 \times \cdots X_K)} = \left\| \sum_{r_2}^{R_2} \left[ \sum_{r_3}^{R_3} \cdots \sum_{r_K}^{R_K} c_{r_1, \cdots, r_K} (\prod_{k=3}^{K} \phi_{r_k}^k) \right] \phi_{r_2}^2 \right\|_{L^2(X_2 \times \cdots X_K)}$$

$$\leq \sum_{r_2}^{R_2} \left\| \sum_{r_3}^{R_3} \cdots \sum_{r_K}^{R_K} c_{r_1, \cdots, r_K} (\prod_{k=3}^{K} \phi_{r_k}^k) \right\|_{L^2(X_2 \times \cdots X_K)} \left\| \phi_{r_2}^2 \right\|_{L^2(X_2)}$$

$$= \sum_{r_2}^{R_2} \left\| \sum_{r_3}^{R_3} \cdots \sum_{r_K}^{R_K} c_{r_1, \cdots, r_K} (\prod_{k=3}^{K} \phi_{r_k}^k) \right\|_{L^2(X_2 \times \cdots X_K)} \tag{23}$$

$$\leq \cdots \leq \sum_{r_2}^{R_2} \cdots \sum_{r_K}^{K} |c_{r_1, \cdots, c_K}|.$$

Substituting Eq.(23) into Eq.(22) and we can get

$$D_1 = \int \sum_{r_1}^{R_1} |C_1|^2 \prod_{k=2}^{K} di_k = \sum_{r_1}^{R_1} \int |C_1|^2 \prod_{k=2}^{K} di_k$$

$$\leq \sum_{r_1}^{R_1} [\sum_{r_2}^{R_2} \cdots \sum_{r_K}^{R_K} |c_{r_1, \cdots, c_K}|]^2 \leq (\prod_{k=2}^{K} R_k) \sum_{r_1}^{R_1} \sum_{r_2}^{R_2} \cdots \sum_{r_K}^{R_K} |c_{r_1, \cdots, c_K}|^2 < (\prod_{k=2}^{K} R_k) \|u\|_{L^2(X_1 \times \cdots X_K)}^2. \tag{24}$$

Next, we have

$$D_2 = \int \sum_{r_1}^{R_1} |C_2|^2 di_1 = \sum_{r_1}^{R_1} \int |\phi_{r_1}^1 - f_{r_1}^1|^2 di_1$$

$$= \sum_{r_1}^{R_1} \left\| \phi_{r_1}^1 - f_{r_1}^1 \right\|_{L^2(X_1)}^2 \tag{25}$$

$$\leq \sum_{r_1=1}^{R_1} [\frac{\epsilon}{(1 + 4 \prod_{k=2}^{K} R_k)^{\frac{r_1}{2}} K \|u\|_{L^2(X_1 \times \cdots \times X_K)}}]^2 < \frac{\epsilon^2}{(4 \prod_{k=2}^{K} R_k) K^2 \|u\|_{L^2(X_1 \times \cdots \times X_K)}^2}.$$

Therefore, we have

$$\|B_1\|_{L^2(X_1 \times \cdots \times X_K)}^2 < D_1 D_2 = (\prod_{k=2}^{K} R_k) \|u\|_{L^2(X_1 \times \cdots \times X_K)}^2 \frac{\epsilon^2}{(4 \prod_{k=2}^{K} R_k) K^2 \|u\|_{L^2(X_1 \times \cdots \times X_K)}^2} = \frac{\epsilon^2}{4K^2}. \tag{26}$$

In the similar sense, we can also prove

$$\|B_k\|_{L^2(X_1 \times \cdots \times X_K)}^2 < \frac{\epsilon^2}{4K^2}. \tag{27}$$

Therefore,

$$A_2 < K \times \sqrt{\frac{\epsilon^2}{4K^2}} = \frac{\epsilon}{2}. \tag{28}$$

Substituting Eq.(14) and Eq.(28) into Eq.(17), we can derive

$$\left\| u - \sum_{r_1}^{R_1} \cdots \sum_{r_K}^{R_K} [c_{r_1,\cdots,r_K} \prod_{k=1}^{K} f_{r_k}^k] \right\|_{L^2(X_1 \times \cdots \times X_K)} \leq A_1 + A_2 < \epsilon. \tag{29}$$

Since $a_{r_1,\cdots,r_k}$ are random coefficients that can take arbitrary values to approximate $c_{r_1,\cdots,r_k}$, we have

$$\left\| u - \sum_{r_1}^{R_1} \cdots \sum_{r_K}^{R_K} [a_{r_1,\cdots,r_K} \prod_{k=1}^{K} f_{r_k}^k] \right\|_{L^2(X_1 \times \cdots \times X_K)} < \epsilon. \tag{30}$$

Thus, we complete the proof.

## B  Illustration of the architecture of the proposed temporally augmented U-Net

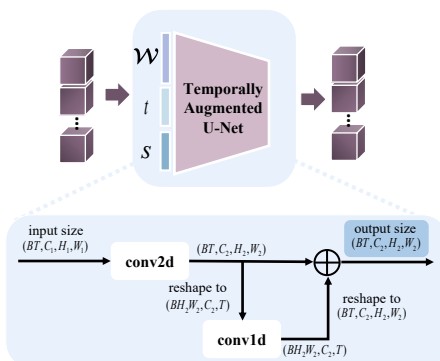

Figure 5: Overview of the Temporally Augmented U-Net. We make a slight modification by adding a Conv1D layer to capture temporal correlations among latent features (A special case of $K = 3$).

We propose a temporally augmented U-Net by incorporating a lightweight temporal enhancement module. As illustrated in Fig. 5, we add Conv1D layers to capture temporal correlations among latent features without altering their original dimensionality. The proposed module requires only minor modifications and is straightforward to implement. Unlike video diffusion models that incur high computational costs by modeling full sequences [41], our approach circumvents this issue by enabling the modeling of frame sequences of arbitrary length and at any desired temporal resolution.

## C  Derivation of guidance gradient of log-likelihood

Here, we derive the guidance gradient of log-likelihood with respect to $\mathcal{W}_{\backslash t_l}^s$:

$$\nabla_{\mathcal{W}_{\backslash t_l}^s} \log p(\mathcal{O}_{t_l} | \mathcal{W}_{\backslash t_l}^s) = \nabla_{\mathcal{W}_{\backslash t_l}^s} \log \int \underbrace{p(\mathcal{O}_{t_l} | \mathcal{W}_{t_l}^0)}_{\text{term 1}} \underbrace{p(\mathcal{W}_{t_l}^0 | \mathcal{W}_{\backslash t_l}^s)}_{\text{term 2}} d\mathcal{W}_{t_l}^0, \tag{31}$$

where $\mathcal{G}_{t_l,t_m}^s = \nabla_{\mathcal{W}_{t_m}^s} \log p(\mathcal{O}_{t_l} | \mathcal{W}_{\backslash t_l}^s)$ is the likelihood gradient of $\mathcal{O}_{t_l}$ respect to $\mathcal{W}_{t_m}^s$, $\mathcal{W}_{\backslash t_l}^s$ denotes perturbed $\mathcal{W}_{\backslash t_l}$ at diffusion step $s$, $\nabla_{\mathcal{W}_{\backslash t_l}^s}$ is the operator to compute the gradient respect to each core in $\mathcal{W}_{\backslash t_l}^s$, and $\mathcal{W}_{t_l}^0$ is the estimated clean target core. Note that in previous step, the latent functions of functional Tucker model are learned. Together with the observation index set specified at $t_l$, the proposed functional Tucker model reduces to a linear model. Therefore, term 1 which corresponding to the likelihood, can be directly expressed as:

$$p(\mathcal{O}_{t_l} | \mathcal{W}_{t_l}^0) \sim \mathcal{N}(\mathbf{y}_{t_l} | \mathbf{A}_{t_l} \mathbf{w}_{t_l}^0, \varepsilon^2 \mathbf{I}), \tag{32}$$

where $\mathbf{y}_{t_l} \in \mathbb{R}^{N_{t_l}}$ is vector containing $N_{t_l}$ entries in $\mathcal{O}_{t_l}$. We collect the indexes of $N_{t_l}$ entries in $\mathcal{O}_{t_l}$ and form the index matrix $\mathbf{I}_{t_l} \in \mathbb{R}^{N_{t_l} \times K}$. Then, $\mathbf{A}_{t_l} \in \mathbb{R}^{N_{t_l} \times \prod_{k=1}^{K} R_k}$ is the factor matrix obtained by evaluating the index matrix $\mathbf{I}_{t_l}$ on the Kronecker product of the learned latent functions, $f_{\theta_1}^1 \otimes \cdots \otimes f_{\theta_K}^K$. $\mathbf{w}_{t_l}^0 \in \mathbb{R}^{\prod_{k=1}^{K} R_k}$ is the vectorized form of $\mathcal{W}_{t_l}^0$.

To make the following derivation clearer, we vectorize each element in $\mathcal{W}_{\backslash t_l}^s$ and concatenate them into a matrix, denoted as $\mathbf{W}_{\backslash t_l}^s \in \mathbb{R}^{M-1 \times \prod_{k=1}^{K} R_k}$. As for term 2, we have

$$
\begin{aligned}
p(\mathcal{W}_{t_l}^0 | \mathcal{W}_{\backslash t_l}^s) = p(\mathbf{w}_{t_l}^0 | \mathbf{W}_s^{\backslash t_l}) &= \int p(\mathbf{w}_{t_l}^0 | \mathbf{W}_{\backslash t_l}^0) p(\mathbf{W}_{\backslash t_l}^0 | \mathbf{W}_{\backslash t_l}^s) \mathrm{d}\mathbf{W}_{\backslash t_l}^0 \\
&= \mathbb{E}_{\mathbf{W}_{\backslash t_l}^0 \sim p(\mathbf{W}_{\backslash t_l}^0 | \mathbf{W}_{\backslash t_l}^s)} [p(\mathbf{w}_{t_l}^0 | \mathbf{W}_{\backslash t_l}^0)].
\end{aligned}
\tag{33}
$$

We approximate the term 2 with

$$
\mathbb{E}_{\mathbf{W}_{\backslash t_l}^0 \sim p(\mathbf{W}_{\backslash t_l}^0 | \mathbf{W}_{\backslash t_l}^s)} [p(\mathbf{w}_{t_l}^0 | \mathbf{W}_{\backslash t_l}^0)] \approx p(\mathbf{w}_{t_l}^0 | \mathbf{W}_{\backslash t_l}^0 = \mathbb{E}[\mathbf{W}_{\backslash t_l}^0 | \mathbf{W}_{\backslash t_l}^s]),
\tag{34}
$$

which closely related to the Jensen's inequality. The approximation error can be quantified with the Jensen gap.

Note that in EDM[2], the denoiser $D_\theta(\cdot)$ can direct approximate the clean cores:

$$
\hat{\mathbf{W}}_{\backslash t_l}^0 = T(\mathcal{W}_{\backslash t_l}^s),
\tag{35}
$$

where $T(\mathcal{W}_{\backslash t_l}^s)$ with shape of $M-1 \times \prod_{k=1}^{K} R_k$ is the concatenation of vectorized estimated clean $\mathcal{W}_{t_m}^0 \in \mathcal{W}_{\backslash t_l}$ from $D_\theta$, which takes every $\mathcal{W}_{t_m}^s \in \mathcal{W}_{\backslash t_l}^s$ as input. Specifically, assume $\mathcal{W}_{t_m}^s$ is the $i$-th element of $\mathcal{W}_{\backslash t_l}^s$, then:

$$
T(\mathcal{W}_{\backslash t_l}^s)(i,:) = \mathrm{vec}(D_\theta(\mathcal{W}_{t_m}^s; \sigma(s))) = \mathrm{vec}(\hat{\mathcal{W}}_{t_m}^0).
\tag{36}
$$

The term 2 now can be estimated using

$$
p(\mathbf{w}_{t_l}^0 | \mathbf{W}_{\backslash t_l}^s) \approx p(\mathbf{w}_{t_l}^0 | T(\mathbf{W}_{\backslash t_l}^s)).
\tag{37}
$$

With similar rationale as in the training process of the GPSD, we impose an element-wise Gaussian process prior over the estimated clean cores $\hat{\mathbf{W}}_0^{\backslash t_l}$. Under this prior, the conditional distribution in (37) is given by:

$$
\begin{aligned}
p(\mathbf{w}_{t_l}^0 \mid \hat{\mathbf{W}}_{\backslash t_l}^0) &\sim \mathcal{N}(\boldsymbol{\mu}_{t_l}, \boldsymbol{\Sigma}_{t_l}), \\
\boldsymbol{\mu}_{t_l} &= \mathbf{k}_{t_l, \mathcal{T}_{\mathrm{tar}}^{\backslash t_l}}^{\mathrm{T}} \mathbf{K}_{\mathcal{T}_{\mathrm{tar}}^{\backslash t_l}, \mathcal{T}_{\mathrm{tar}}^{\backslash t_l}}^{-1} \hat{\mathbf{W}}_0^{\backslash t_l}, \\
\boldsymbol{\Sigma}_{t_l} &= \left( k_{t_l, t_l} - \mathbf{k}_{t_l, \mathcal{T}_{\mathrm{tar}}^{\backslash t_l}}^{\mathrm{T}} \mathbf{K}_{\mathcal{T}_{\mathrm{tar}}^{\backslash t_l}, \mathcal{T}_{\mathrm{tar}}^{\backslash t_l}}^{-1} \mathbf{k}_{\mathcal{T}_{\mathrm{tar}}^{\backslash t_l}, t_l} \right) \mathbf{I},
\end{aligned}
\tag{38}
$$

where $\mathcal{T}_1^{\backslash t_l'}$ denotes the residual target time set, obtained by excluding $t_l$ from $\mathcal{T}_{\mathrm{tar}}$.

Until now, we have

$$
\begin{aligned}
\nabla_{\mathcal{W}_{\backslash t_l}^s} \log p(\mathcal{O}_{t_l} | \mathcal{W}_{\backslash t_l}^s) &= \int p(\mathbf{y}_{t_l} | \mathbf{w}_{t_l}^0; \sigma(s)) p(\mathbf{w}_{t_l}^0 | \mathbf{W}_{\backslash t_l}^s) \mathrm{d}\mathbf{w}_{t_l}^0 \\
&\approx \int p(\mathbf{y}_{t_l} | \mathbf{w}_{t_l}^0; \sigma(s)) p(\mathbf{w}_{t_l}^0 | \hat{\mathbf{W}}_{\backslash t_l}^0) \mathrm{d}\mathbf{w}_{t_l}^0 = E.
\end{aligned}
\tag{39}
$$

By combining (32) and (38), $E$ also has a closed-form distribution:

$$
E \sim \mathcal{N}(\mathbf{y}_{t_l} | \mathbf{A}_{t_l} \boldsymbol{\mu}_{t_l}, \varepsilon^2 \mathbf{I} + \mathbf{A}_{t_l} \boldsymbol{\Sigma}_{t_l} \mathbf{A}_{t_l}^{\mathrm{T}}) = \mathcal{N}(\mathbf{y}_{t_l} | \mathbf{B}_{t_l} T(\mathcal{W}_{\backslash t_l}^s), \tilde{\boldsymbol{\Sigma}}_{t_l}),
\tag{40}
$$

where $\mathbf{B}_{t_1} = \mathbf{A}_{t_l} \mathbf{k}_{t_l, \mathcal{T}_{\mathrm{tar}}^{\backslash t_l}}^{\mathrm{T}} \mathbf{K}_{\mathcal{T}_{\mathrm{tar}}^{\backslash t_l}, \mathcal{T}_{\mathrm{tar}}^{\backslash t_l}}^{-1}$ and $\tilde{\boldsymbol{\Sigma}}_{t_l} = \varepsilon^2 \mathbf{I} + \mathbf{A}_{t_l} \boldsymbol{\Sigma}_{t_l} \mathbf{A}_{t_l}^{\mathrm{T}}$ for abbreviation.

## D   Algorithm

### D.1   Training phase

In our setting, the training data is irregular and sparse observations sampled from $B$ batches of homogeneous physical dynamics at arbitrary timesteps. Given batches of training data, we first use the FTM model to map these irregular points into a shared set of $K$ latent functions and corresponding core sequence batches. The $K$ latent functions are then fixed, and GPSD is used to learn the distribution of the core sequences. Our model is flexible and well-suited for handling irregular physical field data.

## E   Detailed explaination of MPDPS

The primary goal of MPDPS is to exploit the temporal continuity of the core sequence to smoothly propagate guidance from limited observations across the entire sequence. For simplicity, consider the setup in Fig. 2, where we aim to generate the core tensors at three target timesteps: $t_l$, $t'$, and $t_{l+1}$, denoted as $\boldsymbol{\mathcal{W}}_{t_l}$, $\boldsymbol{\mathcal{W}}_{t'}$, and $\boldsymbol{\mathcal{W}}_{t_{l+1}}$, respectively. In this example, observations are only available at $t_l$ and $t_{l+1}$ (i.e., observation timesteps),denoted as $\mathcal{O}_{t_l}$ and $\mathcal{O}_{t_{l+1}}$. Standard DPS methods cannot provide gradient guidance for generating the intermediate core $\boldsymbol{\mathcal{W}}_{t'}$, due to the lack of direct observations at $t'$.

MPDPS addresses this limitation by propagate the information of $\mathcal{O}_{t_l}$ to $\boldsymbol{\mathcal{W}}_{t_l}, \boldsymbol{\mathcal{W}}_{t'}, \boldsymbol{\mathcal{W}}_{t_{l+1}}$ and the information of $\mathcal{O}_{t_{l+1}}$ to $\boldsymbol{\mathcal{W}}_{t_l}, \boldsymbol{\mathcal{W}}_{t'}, \boldsymbol{\mathcal{W}}_{t_{l+1}}$, which means each observation set will contribute to the generation of all target core sequences. Next, we demonstrate how this can be achieved.

Without loss of generality, we take $\mathcal{O}_{t_l}$ as an example. **At diffusion step $s$, we use $\mathcal{O}_{t_l}$ twice to obtain gradient guidance for the three cores**:

**i) Gradient for $\boldsymbol{\mathcal{W}}_{t_l}^s$:** This follows standard DPS. We directly compute the likelihood gradient with respect to $\boldsymbol{\mathcal{W}}_{t_l}^s$ as $\nabla_{\boldsymbol{\mathcal{W}}_{t_l}^s} \log p(\mathcal{O}_{t_l}|\boldsymbol{\mathcal{W}}_{t_l}^s)$, as illustrated by the green & red arrows at $t_l$ in Fig. 2(a).

**ii) Gradient for $\boldsymbol{\mathcal{W}}_{t'}^s$ and $\boldsymbol{\mathcal{W}}_{t_{l+1}}^s$:**

- Compute denoised cores at $t'$ and $t_{l+1}$: we first use $\boldsymbol{\mathcal{W}}_{t'}^s$ and $\boldsymbol{\mathcal{W}}_{t_{l+1}}^s$ to estimate the corrsponding denoised cores $\hat{\boldsymbol{\mathcal{W}}}_{t'}^0 = D_\theta(\boldsymbol{\mathcal{W}}_{t'}^s)$ and $\hat{\boldsymbol{\mathcal{W}}}_{t_{l+1}}^0 = D_\theta(\boldsymbol{\mathcal{W}}_{t+1}^s)$ via the pretrained denoiser $D_\theta(\cdot)$.

- Estimate pseudo cores at $t_l$ via GPR: With the temporal continuity assumption, we use Gaussian Process Regression (GPR) to estimate the pseudo denoised core $\hat{\boldsymbol{\mathcal{W}}}_{t_l}^0 = \text{GPR}(\hat{\boldsymbol{\mathcal{W}}}_{t'}^0, \hat{\boldsymbol{\mathcal{W}}}_{t_{l+1}}^0) = \text{GPR}(D_\theta(\boldsymbol{\mathcal{W}}_{t'}^s), D_\theta(\boldsymbol{\mathcal{W}}_{t+1}^s))$, which can be viewed as a regression-based prediction using two denoised cores.

- Gradient computation: With pseudo core $\hat{\boldsymbol{\mathcal{W}}}_{t_l}^0$, we compute an additional likelihood of $\mathcal{O}_{t_l}$. Then we can then compute the gradient of the likelihood respect to $\boldsymbol{\mathcal{W}}_{t'}^s$ and $\boldsymbol{\mathcal{W}}_{t_{l+1}}^s$, as they directly determine $\hat{\boldsymbol{\mathcal{W}}}_{t_l}^0$ via $D_\theta$ and GPR. We further derive a **closed-form solution** in Appendix C for gradients with respect to $\boldsymbol{\mathcal{W}}_{t'}^s$ and $\boldsymbol{\mathcal{W}}_{t_{l+1}}^s$ (i.e., $\nabla_{\mathcal{W}_{\backslash t_l}^s} \log p(\mathcal{O}_{t_l}|\mathcal{W}_{\backslash t_l}^s)$), which can be efficiently computed. The results are shown in Eq. 1011.

The process is illustrated by the green arrows pointing to $\mathcal{O}_{t_l}$ in Fig. 2(b).

Similarly, information from $\mathcal{O}_{t_{l+1}}$ is propagated to $\boldsymbol{\mathcal{W}}_{t_l}, \boldsymbol{\mathcal{W}}_{t'}$ and $\boldsymbol{\mathcal{W}}_{t_{l+1}}$ in the same manner, and we sum the gradients from all observations to obtain the final gradient guidance for each core, as shown in Eq. 12.

In this way, each observation contributes to the generation of all target timesteps. In other words, every target coreregardless of whether it has a corresponding observationreceives information from all observed timesteps. To sample the entire core sequence, we simply add the gradients from all observations to the posterior, as shown in Eq. 12. This mechanism encourages smooth and coherent generation across the entire sequence.

# F Implementation details

All the methods are implemented with PyTorch [42] and trained using Adam [43] optimizer with the learning rate tuned from $\{5e^{-4}, 1e^{-3}, 5e^{-3}, 1e^{-2}\}$.

For DEMOTE, we used two hidden layers for both the reaction process and entry value prediction, with the layer width chosen from $\{128, 256, 512\}$. For LRTFR, we used two hidden layers with layer width chosen from $\{128, 256, 512\}$ to parameterize the latent function of each mode. We varied $R$ from $\{16, 32\}$ for all baselines. For Senseiver, we used 128 channels in both the encoder and decoder, a sequence size of 256 for the $Q_{in}$ array, and set the size of the linear layers in the encoder and decoder to 128. For CoNFiLD, we used a Conditional Neural Field module with a latent dimension of 256 for the *Ocean Sound Speed* and *Active Matter* datasets, and 1024 for the *Supernova Explosion* dataset. The diffusion model module was configured with 100 sampling steps. For our method, we first apply a functional Tucker model to decompose the tensor into factor functions and a core sequence. Each factor function is parameterized by a three-layer MLP, where each layer contains 1024 neurons and uses the sine activation function. The core sizes are set to $32 \times 32 \times 32$, $3 \times 12 \times 12$, and $48 \times 48$ for the *Supernova Explosion*, *Ocean Sound Speed*, and *Active Matter* datasets, respectively. These hyperparameters are carefully selected to achieve optimal performance.

## F.1 Defination of Variance-scaled Root Mean Squared Error

Let $\{\hat{y}_i\}_{i=1}^N$ and $\{y_i\}_{i=1}^N$ denote the predicted and ground-truth entry, respectively. Assume that there are $N$ points in total. The Variance-scaled Root Mean Squared Error (VRMSE) is defined as

$$\text{VRMSE} = \frac{\sqrt{\frac{1}{N}\sum_{i=1}^N (\hat{y}_i - y_i)^2}}{\sqrt{\frac{1}{N}\sum_{i=1}^N (y_i - \bar{y})^2}}, \tag{41}$$

where $\bar{y}$ is the mean of all points.

## F.2 Guided Sampling Details

|  | Supernova Explosion | Ocean Sound Speed | Active Matter |
|---|---|---|---|
| $\zeta_{\text{SDIFT}}$ | $1 \times 10^{-2}$ | $3 \times 10^{-2}$ | $1 \times 10^{-2}$ |
| $\zeta_{\text{CoNFiLD}}$ | $1 \times 10^{-1}$ | $1 \times 10^{-1}$ | $1.2 \times 10^{-1}$ |

Table 3: Weights assigned to posterior guidance by CoNFiLD and SDIFT.

For experiments conducted on three datasets with sparse observations, we use the weights $\zeta$ in Tab. 3.

# G Additional experiment results

## G.1 Unconditional generation

In this subsection, we present unconditional generation results on three datasets of SDIFT, demonstrating its ability to generate physical fields from irregular and sparse training data. The unconditional generation results on *Supernova Explosion* are shown in Fig. 6; The unconditional generation results on *Ocean Sound Speed* are shown in Fig. 7; The unconditional generation results on *Active Matter* are shown in Fig. 8. **One can see that the unconditional generations are of high quality and smooth across frames, demonstrating the effectiveness of SDIFT in generating full-field physical dynamics from irregular sparse observations.**

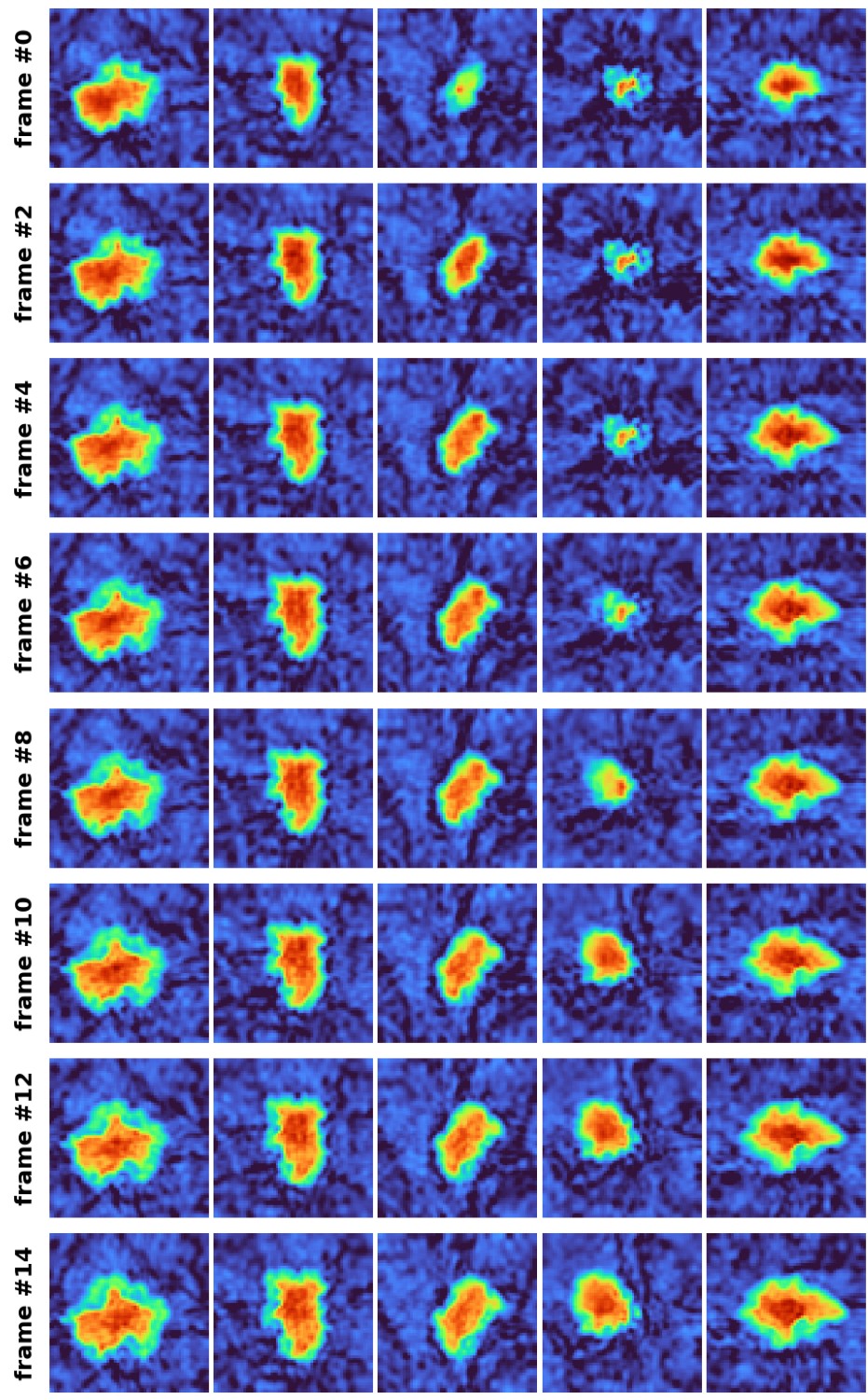

Figure 6: Unconditional generation results on *Supernova Explosion* by SDIFT.

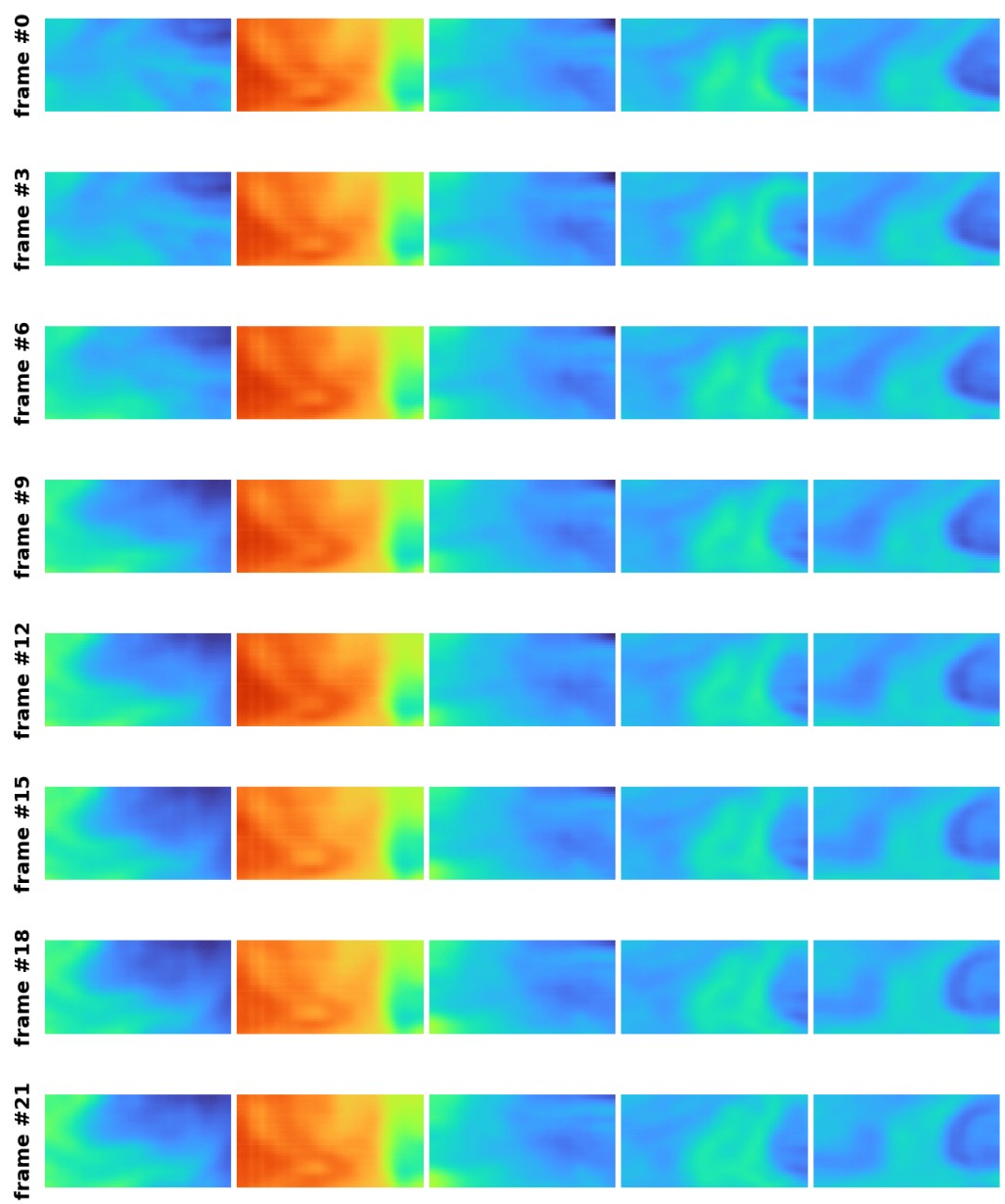

Figure 7: Unconditional generation results on *Ocean Sound Speed* by SDIFT.

## G.2 More reconstruction results

More visual results of three datasets conducted on observation setting 1 are illustrated in Fig. 91011.

More visual results of three datasets conducted on observation setting 2 are illustrated in Fig. 121314.

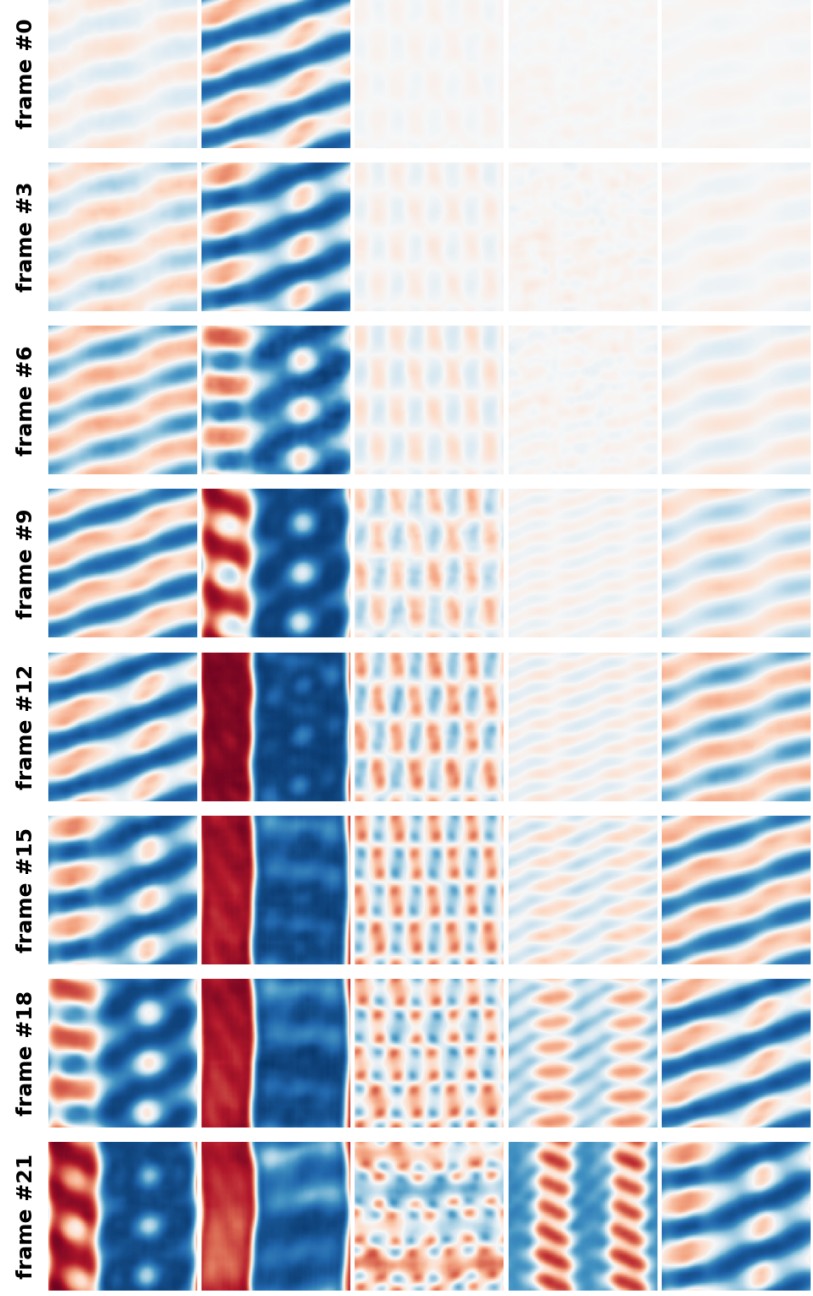

Figure 8: Unconditional generation results on *Activate Matter* by SDIFT.

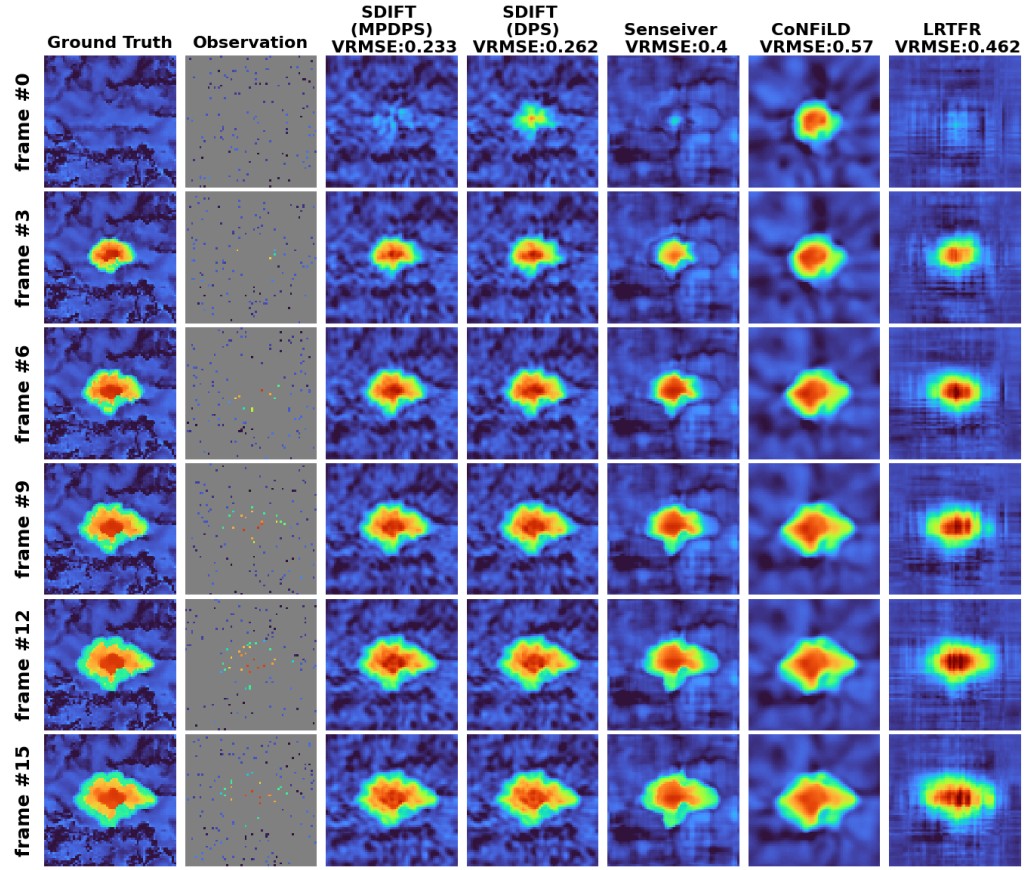

Figure 9: Visual reconstruction results of *Supernova Explosion* dynamics under **observation setting 1** with $\rho = 1\%$.

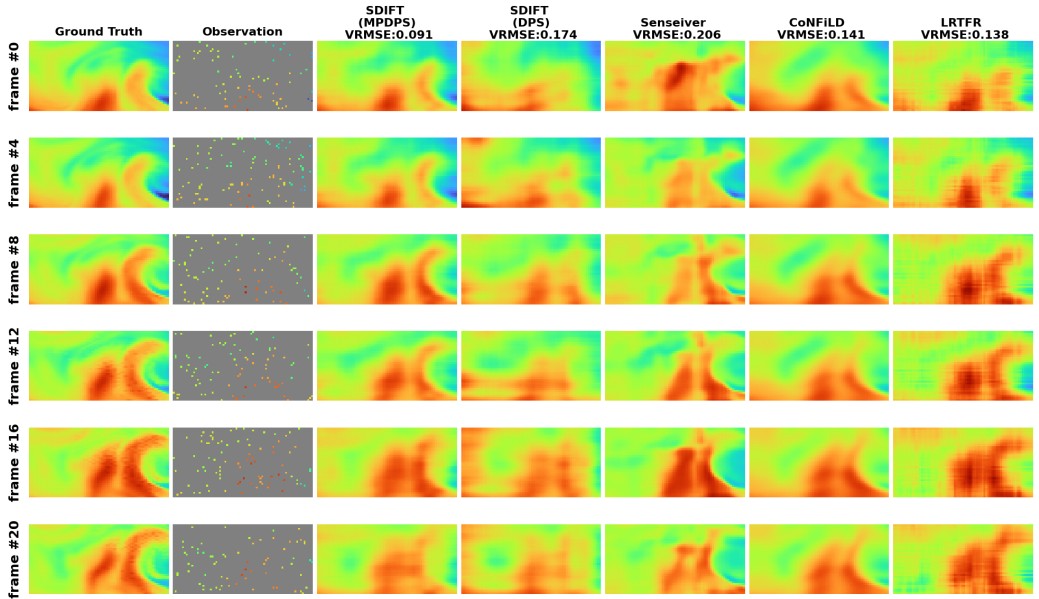

Figure 10: Visual reconstruction results of *Ocean Sound Speed* dynamics under **observation setting 1** with $\rho = 3\%$.

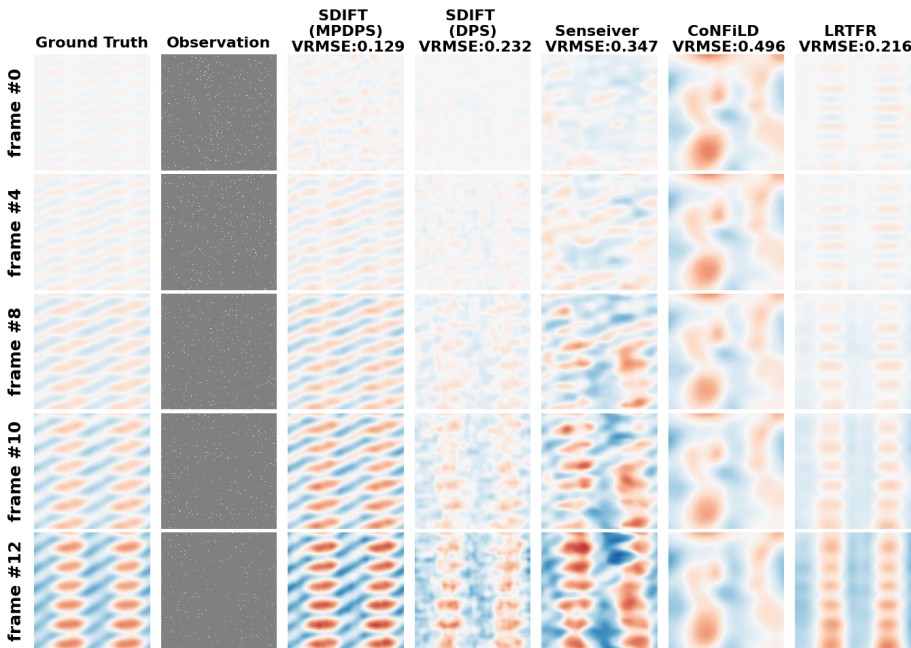

Figure 11: Visual reconstruction results of *Active Matter* dynamics under **observation setting 1** with $\rho = 1\%$.

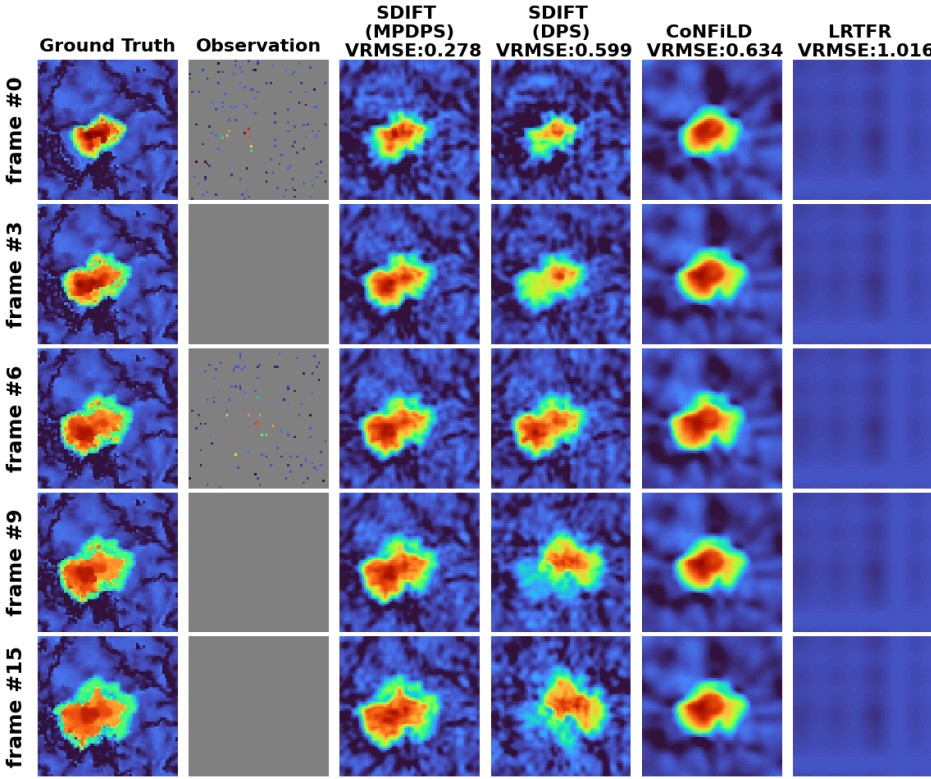

Figure 12: Visual reconstruction results of *Supernova Explosion* dynamics under **observation setting 2** with $\rho = 3\%$.

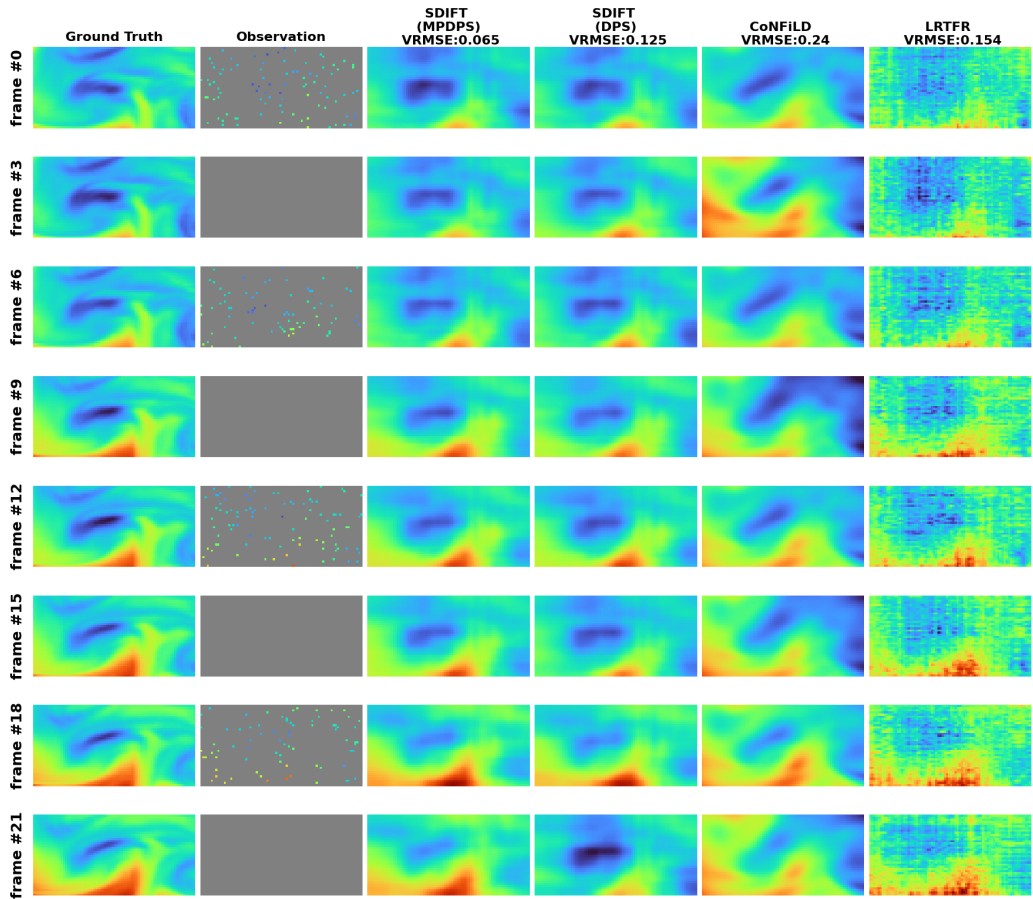

Figure 13: Visual reconstruction results of *Ocean Sound Speed* dynamics under **observation setting 2** with $\rho = 3\%$.

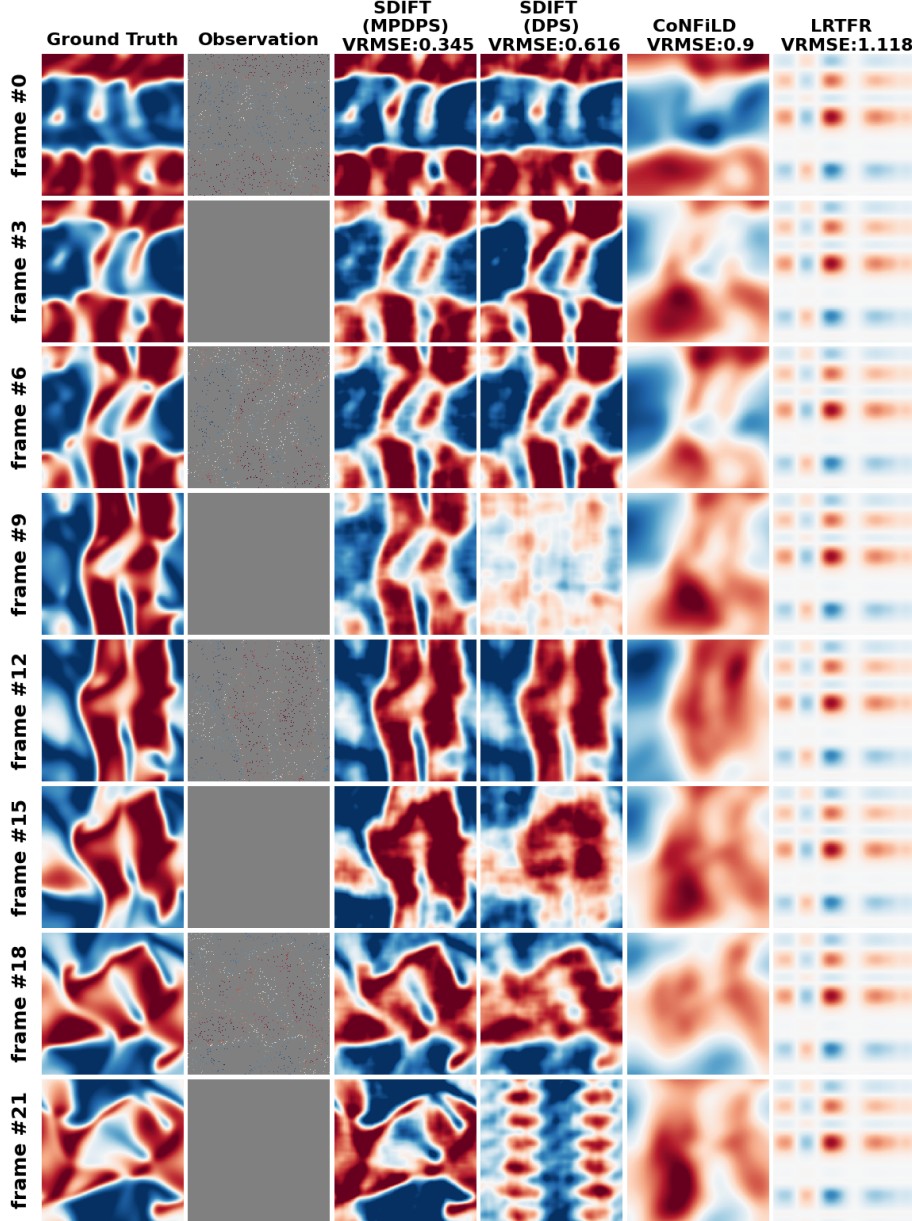

Figure 14: Visual reconstruction results of *Active Matter* dynamics under **observation setting 2** with $\rho = 3\%$. Since DPS does not provide guidance at timesteps without observations (**i.e., frame # 3,9,15,21**), SDIFT generates the corresponding physical fields randomly. In contrast, MPDPS effectively overcomes this limitation and produces smooth reconstructions. This phenomenon highlights the significance of the proposed MPDPS.

### G.3 Analysis on the results of CoNFiLD

CoNFiLD [5] is closely related to our approach. It first uses conditional neural fields to learn latent vectors for off-grid physical field data over time. Next, it simple concatenates the $T$ latent vectors extracted from a record into a latent matrix and treats this matrix as an image, applying a standard diffusion model to learn its distribution. We present dimensionality reduction visualizations of the latent vectors and core tensors extracted from *Active Matter* dataset using CNF and our FTM, respectively, in Fig. 15(a)(b). We use points with the same color hue to represent representations from the same record, and vary the color intensity to indicate different timesteps. We plot 9 records, with each records containing 24 timesteps. One can see that both kinds of representation show strong time continuity.

CoNFiLD use traditional UNet to learn the distribution of latent matrices (i.e., the sequence of scatter points in Fig. 15(a)), which is hard to capture complex temporal dynamics (due to the inductive bias of CNN[44]). Our experiments demonstrate that, although CNF produces accurate reconstructions from its latent representations, **the resulting latent matrices are difficult to model with standard diffusion approachesespecially when the physical field changes rapidly, as in the *Active Matter* datasetleading to poor posterior sampling performance.** On the *Ocean Sound Speed* dataset, where the field varies much more slowly and smoothly, the diffusion model performs significantly better.

In contrast, we propose a temporally augmented U-Net that learns the distribution of each core point (i.e., the individual points in Fig. 15(b)) while capturing temporal correlations through the Conv1D module. This approach significantly simplifies modeling the full latent distribution, whether it changes rapidly or slowly. Furthermore, our approach can flexibly generate sequences of arbitrary length with arbitrary time intervals, whereas CoNFiLD cannot.

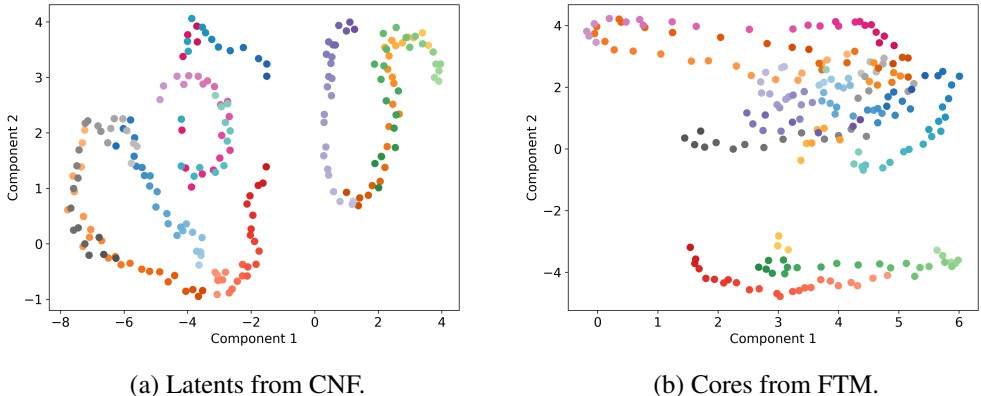

(a) Latents from CNF.    (b) Cores from FTM.

Figure 15: Illustration of dimensionality reduction of the latent vectors and core tensors extracted from *Active Matter* using CNF and our FTM, receptively.

### G.4 Robustness against noise

We demonstrate the robustness of our method against different levels and different types of noise in Table 4.

### G.5 Ablation study on using GP noise as the diffusion source

We conduct an ablation study on the use of GP noise as the diffusion source by replacing it with i.i.d. Gaussian noise, using the *Active Matter* dataset as an example.

Fig. 16 shows unconditional generations from our method when i.i.d. Gaussian noise is used as the diffusion source. Compared to Fig. 8, this replacement significantly disrupts temporal continuity and degrades generation quality.

| Noise configurations | SDIFT w/ DPS $\rho = 1\%$ | SDIFT w/ MPDPS $\rho = 1\%$ |
|---|---|---|
| Gaussian noise ($\sigma = 0.1$) | $0.207 \pm 0.066$ | $\mathbf{0.156 \pm 0.052}$ |
| Gaussian noise ($\sigma = 0.3$) | $0.210 \pm 0.069$ | $\mathbf{0.164 \pm 0.049}$ |
| Laplacian noise ($\sigma = 0.1$) | $0.200 \pm 0.075$ | $\mathbf{0.170 \pm 0.054}$ |
| Laplacian noise ($\sigma = 0.3$) | $0.224 \pm 0.054$ | $\mathbf{0.177 \pm 0.051}$ |
| Poisson noise ($\sigma = 0.1$) | $0.186 \pm 0.081$ | $\mathbf{0.168 \pm 0.048}$ |
| Poisson noise ($\sigma = 0.3$) | $0.214 \pm 0.070$ | $\mathbf{0.171 \pm 0.053}$ |

Table 4: VRMSE of reconstruction of our proposed method on *Ocean Sound Speed* datasets over varying noise with observation setting 1 and $\rho = 1\%$.

| VRMSE | observation setting 1 $\rho = 1\%$ | observation setting 2 $\rho = 1\%$ |
|---|---|---|
| GP noise($\gamma = 50$) | $0.215 \pm 0.068$ | $0.296 \pm 0.096$ |
| GP noise($\gamma = 100$) | $0.222 \pm 0.077$ | $0.304 \pm 0.104$ |
| GP noise($\gamma = 200$) | $0.219 \pm 0.084$ | $0.298 \pm 0.084$ |
| i.i.d Gaussian noise | $0.257 \pm 0.079$ | $0.393 \pm 0.109$ |

Table 5: VRMSE on *Active Matter* dataset over different noise source using SDIFT guided by MPDPS.

We also report the VRMSE of reconstruction results in Table 5. Replacing GP noise with i.i.d. Gaussian noise as the diffusion source markedly worsens performance. In contrast, GP noise yields consistently low VRMSE and is robust to variations in the kernel hyperparameter $\gamma$, further demonstrating its suitability as a diffusion source in our settings.

# H   Limitations

A current limitation of our work is the lack of explicit incorporation of physical laws into the modeling process. In addition, the introduced Gaussian Process prior relies on the choice of kernel functions; in this work, we only use the simple RBF kernel as an example and do not explore other options in detail.

In future work, we plan to integrate SDIFT with domain-specific physical knowledge to enable more accurate long-range and wide-area reconstructions of physical fields. We also intend to investigate a broader range of kernel functions to enhance modeling flexibility and performance.

# I   Impact Statement

This paper focuses on advancing generative modeling techniques for physical fields. We are mindful of the broader ethical implications associated with technological progress in this field. Although immediate societal impacts may not be evident, we recognize the importance of maintaining ongoing vigilance regarding the ethical use of these advancements. It is crucial to continuously evaluate and address potential implications to ensure responsible development and application in diverse scenarios.

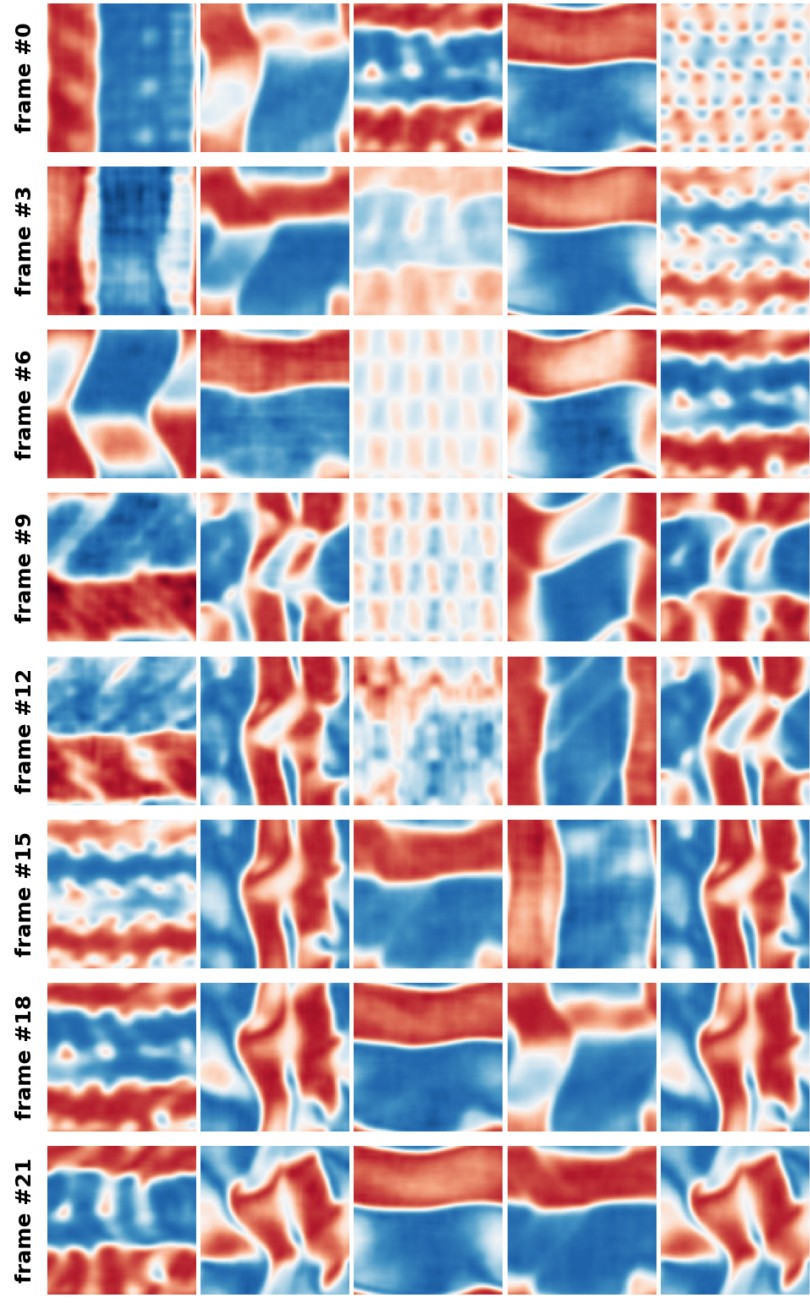

Figure 16: Ablation study: Unconditional generation results on *Activate Matter* by SDIFT using **i.i.d Gaussian noise** as the diffusion source. Note that this replacement significantly disrupts temporal continuity and degrades generation quality.

