# OpenReview forum: "Generating Full-field Evolution of Physical Dynamics from Irregular Sparse Observations"
_NeurIPS.cc/2025/Conference — NeurIPS 2025 poster_

### Official Review · Reviewer_BZPv · 2025-06-13

**Clarity:** 1
**Significance:** 2
**Originality:** 3
**Rating:** 3
**Confidence:** 3

**Summary:**

The paper proposes a latent diffusion model for spatiotemporal data, where the latent representation of each temporal snapshot follows a Tucker decomposition, and diffusion learns the distribution of Tucker factors. The paper proposes message-passing to denoise temporally consistent frames. The method works very well on interpolating frames from very sparse data.

**Questions:**

- The role of modes could be clarified. The paper starts with talking about K modes, which I think are just the spatial coordinates (ie. x,y,z in 3D data). Do we ever have K>3?
- Is time one of the modes? Why or why not?
- What does “solving” eq 6 mean in line 149? One needs here both the W and f’s: are both optimised? How do you optimise? Having an algorithm box would be helpful.
- The remark about VAE is rather strange: I don’t see anything variational, stochastic or Bayesian around eq 6.
- Line 163 claims interpretable cores. I don’t see any demonstration of this: how are the cores interpretable?
- What is “alternating-direction strategy”?
- It seems that one only learns one scalar function per dimension (line 170). This feels quite limited: I wonder what the f-core is then learning or representing? What do the f-cores look like in practise?
- Is learning of W and f identifiable? One would expect that some rotational invariances would exist. Is their learning sensitive to initialisation? In line 176 the paper assumes that W_i equals W(t_i), but I don’t think you can state this unless the encoding of W_i is deterministic, which requires identifiability and invariance to initial values. In practise the encoding seems stochastic, and one would then have some stochasticity in the W_i’s you get. I also wonder what “drawn from a continuous function” means. We draw samples from a distribution, but evaluate a function. Is there some stochastic process interpretation here? Can you discuss?
- In line 178 the W(t) is remarked as distribution. Why is it a distribution?
- Can you clarify the roles of s and t in eq 9. If t is an observation time, then why not also integrate over observation locations?
- I’m a bit confused what \Eps or \W look like: what are their dimensions? Line 184 says that \Eps is only a function of time, but it is added to \W which only contains spatial stuff. Is the idea that Eps-set is kept fixed? Or is there some kind of Eps-minibatching?
- Line 200 says that we can sample using eq 9 model for any (i,t). Does eq 9 contain i?
- In line 204 it is stated that observations are a subset of target times. Why do we want to predict target times if we already observe them? Can you clarify what is test and train?
- I can’t follow eqs 10-12
- How does the train/test split work in experiments? Do you do interpolation or forecasting? How does this relate to the Tar/obs splitting? It seems that half of target is obs, but 95% of frames are train and 5% are test. I’m not sure I understand what does target/test/obs/train mean.
- How do you use the model at training and at testing stage?

**Ethical Concerns:**

["NO or VERY MINOR ethics concerns only"]

**Final Justification:**

The response was informative and resolved my confusions about the work. This is a solid work with great performance and good novel method contribution, but the clarity and wider ML significance are still weaknesses of the work. I'm leaning on borderline rejection, but would not object to acceptance.

**Limitations:**

No issues

**Paper Formatting Concerns:**

There might be some compression tricks going on. Fig4 and Table 1 captions are very close to succeeding text; and sec 3 title does not seem to have normal amount of margins.

**Quality:**

2

**Strengths And Weaknesses:**

Originality good. The paper combines existing ideas, but the use of Tucker as diffusion latent representation is innovative and a good match for the domain problem.

Significance mixed. The scope of the paper is physics-based, low-resolution spatiotemporal dynamics, where some frames are missing and remaining ones have 1-3% of the pixels observed. The task is to impute the field. The task is interesting and is a great stress-test challenge for the methods. However, the task is also very specific, and the paper does not demonstrate why it’s a significant or realistic application. I'm not sure if the paper demonstrates "real-world" usecases. The paper does not discuss if the contributions have more general ML relevance.

Quality mixed. The method has good and consistent performance. The method is well-motivated and sound. The main issues of quality are in weak clarity.

Clarity weak. The method is relatively simple (denoising latent factors), but the notation is overly complex. The paper defines a datapoint as (i_n_m, t_m, y_i_n_m), and most parts of the paper inherit this complexity which makes the paper hard to read and digest. The K-mode notation seems unnecessary since all experiments are in 3D or 2D. The motivation, open problem and setting are vague [is this a PDE paper? What is “off-grid”? Do we interpolate or extrapolate or forecast?], and the paper does not clearly position itself to the literature. The message-passing sections are borderline unreadable to the the complicated notation. The method is a black-box, and little insight is given how the model denoises, or why (visualisations would help). There are no ablations showing how the different terms improve (Tucker vs other latent representations; message-passing vs other temporal consistency guidance methods, etc). The paper needs to dramatically improve it’s notation, and borrow good practises from exemplar diffusion papers.

Overall the paper is an ok technical report showing a succesful method, but as a scientific paper it needs to demonstrate the ideas and significance in wider ML context, and improve presentation.

---

> ### Author Rebuttal · Authors · 2025-07-30
>
> We thank the reviewer for the careful review and  will improve the clarity in the latest version.  We address the comments below:(C: comment; R: response)
>
> To aid the reviewer’s understanding, we first claim our paper’s main objective, basic concepts, and experiment setup:
> |     |  |
> |:-----:|:-----|
> |**main objective & real-world applications** | The **main objective** of this paper is to  reconstruct complex time-evolving physical fields from sparse sensor observations. This scientific task is   challenging yet important  across diverse domains, including industry, medicine, and science. Applications range from structural health monitoring, tectonic motion estimation,  climate modeling, and so on [18]. This problem has attracted extensive attention, including in leading journals such as *Nature* [5,6,18]. |
> |**Basic concepts** | **1.Off-grid data** are irregularly sampled, sparse measurements from sensors distributed unevenly in space and time, which cannot be represented as regular grids or tensors. **2.Functional Tucker model for representation**:We use FTM as a universal representor to map each snapshot’s datapoints into a core tensor and $K$ shared latent factor functions. The latent factors are shared across snapshots, while the cores vary. For computational details, pls refer to R2 of Reviewer dp1M.The learned latent functions serve as positional basis functions for each mode, with the core tensor providing their weights. *This formulation decouples the temporal mode (via the core) from spatial or other modes (via the latent factors)*.Thus, we can focus on modeling temporal dynamics by learning the distribution of core sequences using a diffusion model.**3. Latent diffusion model** efficiently maps data (i.e., images or video frames) into a low‑dimensional latent space to perform noise‑based generation and editing,  improving efficiency while preserving generation quality. It generate latents by iteratively denoising a noisy input through a learned reverse process.|
> |**Experiment setup**| Assume we have a total of $B$ physical dynamics, split into a training set with $B_1$ and a test set with $B_2$ dynamics, where $B_1 + B_2 = B$ and the two sets are disjoint. To simulate realistic scenarios where sensor data is inherently sparse and irregular, we randomly mask out 80% of the data in the $B_1$ set, using the remaining 20% to train FTM and GPSD.During testing, we sample only 1% or 3% of the data from the $B_2$ set to reflect practical situations with highly sparse observations. These sparse samples are used to guide reconstruction via gradient updates, using the pre-trained FTM and GPSD. The reconstructed full fields are then compared against the ground-truth $B_2$ dynamics.*In conclusion, our method is able to handle irregular data both in training phase and test phase.* |
>
> >C1: Confusion on more general ML relevance
>
> R1: We propose a new generative model that directly operates on off-grid multidimensional data during training and testing phase, which is broadly applicable to unstructured domains like sensor networks, point clouds, and scientific measurements.
>
>
>
> >C2: Concerns on (i_n_m, t_m, y_i_n_m)  and K-mode notation
>
> R2:We clarify that this work aims to construct a generative model directly from **off-grid** data, which cannot be represented as regular matrices or tensors. Instead, we treat the data as points, following common practice in the literature [12, 31].
>
> This work builds a generative model from **off-grid** data, treated as point sets rather than regular tensors, following common practice [12, 31].
>
> The value of $K$ is for notational rigor and is will be greater than 3 in practice—for example, $K = 4$ for 3D spatial + frequency radio map/seismic data, or $K = 5$ for multispectral data with 3D spatial, frequency and observation angular modes. Our experiments focus on 2D and 3D cases for simplicity.
>
>
>
> >C3: Question on the time mode
>
> R3:The temporal mode is not included in $K$. The reason is that the temporal mode usually exhibits much sharper or more complex fluctuations that cannot be effectively captured by a single decoupled  latent factor function [31].Instead, we treat it separately using our proposed GPSD module, which leverages Gaussian processes and a temporally-augmented UNet to model temporal correlations.
>
> >C4:What does “solving” eq 6 mean?What is “alternating-direction strategy”?
>
> R4: Solving Eq. (6) means learning the mapping between data and a low-dimensional latent space by minimizing Eq. (8) using an alternating optimization that updates $\mathcal{W}$ and $f_{\theta}$ parameters in turn.
>
> >C5:The remark about VAE is rather strange
>
> R5: We clarify that FTM is similar to VAEs in function—mapping data to a latent space—rather than in specific techniques.
>
>
> >C6: Visualisation for  denoisng  cores.
>
> R6: We will  add visualizations in the latest version.
>
> >C7: Ablations on Tucker vs other latent representations; message-passing vs other temporal consistency guidance methods.
>
> R7:We use FTM as the latent representation to learn a  mapping from irregular multidimensional physical fields to a lower-dimensional space, as other latent representations like VAE cannot  handle such irregular data in our setting.
>
> To the best of  our knowledge, **we are the first to introduce temporal consistency for diffusion guidance.** However, we can use  common interpolation methods like linear or spline to interpolate gradients for those timesteps without observations.  We conduct experiments on  Observation setting 2 with three datasets. The results are:
> |     | MPDPS | DPS+spline|  DPS+linear|
> |:-----:|:-----:|:-----:|:-----:|
> | Supernova Explosion ($\rho=$1%) |**0.433$\pm$0.163** |0.581$\pm$0.298 |0.621$\pm$0.227|
> | Ocean Sound Speed ($\rho=$1%) |  **0.181$\pm$0.084** | 0.294$\pm$0.12 | 0.342$\pm$0.181|
> | Active Matter ($\rho=$1%) |  **0.296$\pm$0.096** | 0.487$\pm$0.210 | 0.512$\pm$0.233|
>
>  One can see that  our MPDPS performs much better. We will supplement these into the latest version.
>
>
> >C8: Missunderstanding on factor function
>
> R8: We kindly refer you to line 146, where we clarify the latent function as **a vector-valued function.** The learned latent functions can be interpreted as positional bases functions for each mode, while the core tensor serves as the weighting coefficients that interact with these bases across the $K$ modes.
>
> >C9: Interpretion of core
>
> R9:The core tensor serves as weighting coefficients for the $K$-mode bases, summarizing the full field at each snapshot. Pls see Sec.6 of [22] for  details of its structural patterns.
>
> >C10:Concern on identifiability, rotational invariance, and sensitivity to initialisation
>
> R10:
> - FTM does not produce identifiable $\mathcal{W}$ and $f$ due to the freedom of invertible transformations, as you noted, but this is not an issue since our reconstruction task—unlike blind source separation—does not require identifiability. Like deep matrix factorization and VAEs, these transformations leave the reconstruction loss unchanged. Once $f$ is learned and fixed, the rotational ambiguity of $\mathcal{W}$ is also fixed during diffusion training and sampling.
>
> - For the sensitivity to initialisation, we acknowledge the initialisation is crucial for model training, just like most deep-based models' training. In practice, we train with Kaiming initialisation, AdamW, and (mini‑)batch gradient descent on the convex quadratic decoder of Eq. (6).Across 5 random seeds the VRMSE standard deviation is < $10^{-3}$, showing that the model converges stably regardless of the initial values.We will add this table to the appendix in the latest version.
>
> >C11:question about meaning of “drawn from a continuous function $\mathcal W(t)$ and stochastic process interpretation”
>
> R11:
> - On the **function** view, we regard the core trajectory as a continuous map $\mathcal W:\mathbb R^{+}\to\mathbb R^{R_1\times\cdots\times R_K}$. meaning that it can be evaluated at any time point $t$ within the range of interest. The observed cores $\mathcal W_{t_m}$ are simply samples of this function at discrete time stamps, hence the phrase “drawn”.
> - On the **stochastic‑process** view, in the generative (Stage‑2) part we place an element‑wise Gaussian‑process (GP) prior on $\mathcal W(t)$ and inject GP noise in the forward diffusion.  The reverse process denoises these stochastic samples.  Thus, in training, we learn the GP prior and the denoiser to model the temporal correlations of the core sequence. This is similar to how standard diffusion models learn the distribution of images or videos in pixel space, but here we do it in a latent function space. In the sampling (Stage‑3) part, we sample the core sequence from the learned GP prior and denoise it to obtain the final core sequence.
>
> >C12:Confusion about the componants of diffusion model (s,t,i)
>
> R12: s:diffusion step; t:time step.  In Stage 1, we use FTM to map all data points at time step $t$ into a Tucker core $\mathcal{W}_t$. The spatial information is captured by latent factor functions and is decoupled from $\mathcal{W}_t$. Thus, $\mathcal{W}_t$ summarizes the full-field information at time $t$ and only the temporal mode needs to be modeled during diffusion training.
>
>
> >C13: Why do we want to predict target times if we already observe them?
>
>
> R13: Our main task is to reconstruct time-evolving **full fields** from sensor observations. Although the target time points have some sparse observations, they are far from enough for downstream tasks. Hence, completing the entire field is necessary.
>
>
> >C14: I can’t follow eqs 10-12
>
> R14:  For  detailed explanations, pls refer to R5 of Reviewer wHTh.
>
> >C15: Questions on target/test/obs/train
>
> R15: Pls refer to the experiment setup at the begining for the train/test split. In the test phase, the **target time** refers to the timesteps where we aim to reconstruct the full field, while the **observation time** refers to the timesteps where sparse observations are available.

---

> > ### Comment · Reviewer_BZPv · 2025-08-04
> >
> > Thanks for the informative response. My confidence in the work has increased, and I'm increasing my score to 3.

---

> > > ### Author Response · Authors · 2025-08-04
> > >
> > > Thank you very much for your time and thoughtful comments.  Please feel free to reach out with any further questions or suggestions — we would be happy to address them at any time.

---

> ### Author Response · Authors · 2025-08-08
> **Gentle Request for Further Discussion**
>
> Given that the score is slightly below the borderline, we would like to know  what other steps we can take to further improve our paper or address your concerns？

---

### Official Review · Reviewer_uVzD · 2025-06-25

**Clarity:** 4
**Significance:** 4
**Originality:** 3
**Rating:** 5
**Confidence:** 2

**Summary:**

This paper tackles an important and practically relevant problem—reconstructing full-field spatiotemporal physical dynamics from irregular and sparse observations. The proposed SDIFT framework is methodologically innovative and theoretically sound. It builds on the Functional Tucker Model as a continuous latent representation with proven universal approximation capability, and further integrates a Gaussian process–based diffusion mechanism and a novel message-passing posterior sampling strategy. The technical contributions are solid, well-motivated, and clearly described, with thorough empirical validation across diverse physical systems. Overall, the paper offers a compelling and elegant solution to a challenging generative modeling task.

**Questions:**

1. Since the method does not require pretraining on densely sampled dynamics, I am curious about its behavior under extremely sparse observation conditions. Specifically:

- As the number of observations decreases, is there a threshold below which the model fails to generate meaningful reconstructions due to a lack of information?
- How does the model perform under non-uniform observation patterns, such as those common in weather-related applications, where sensors are densely located in developed regions but sparse over oceans or remote areas? This setting might challenge the assumption of homogeneity in sampling patterns.

2. In some domain-specific applications, methods have been proposed to perform field reconstruction without pretraining. For example, [1] uses a learnable AI model to represent the evolution process and directly minimizes a loss function that enforces consistency between the predicted and observed values. This formulation seems more amenable to incorporating physical constraints or domain priors. Could you clarify the main advantages of the method compared to such approaches, particularly in terms of flexibility, generalizability, or computational efficiency?

[1] Fablet, Ronan, Bertrand Chapron, Lucas Drumetz, Etienne Mémin, Olivier Pannekoucke, and François Rousseau. "Learning variational data assimilation models and solvers." *Journal of Advances in Modeling Earth Systems* 13, no. 10 (2021): e2021MS002572.

**Ethical Concerns:**

["NO or VERY MINOR ethics concerns only"]

**Final Justification:**

The paper tackles the highly non-trivial task of reconstructing full-field physical dynamics from sparse, irregular, and off-grid observations, by proposing a framework which supported by a solid theoretical foundation. From my perspective, the method is interesting and the overall design appears reasonable and well-motivated. I acknowledge that I am not deeply familiar with the literature on tensor-based representations for physical field modeling, so I may have missed potential limitations in that aspect of the method and can't really evaluate its novelty.

**Limitations:**

Yes, the authors have discussed the limitations.

**Quality:**

4

**Strengths And Weaknesses:**

Strengths:

1. Challenging and important problem setting: The paper tackles the highly non-trivial task of reconstructing full-field physical dynamics from sparse, irregular, and off-grid observations. Unlike many existing approaches that require access to dense spatiotemporal trajectories for pretraining, SDIFT is capable of learning directly from sparse observations, making it more practical and broadly applicable in real-world scenarios.
2. Theoretical soundness: The proposed framework is supported by a solid theoretical foundation. In particular, the authors rigorously prove the universal approximation property of the Functional Tucker Model (FTM).
3. Comprehensive experiments: The paper presents extensive experimental validation across three distinct physical systems (supernova explosions, ocean sound speed, and active matter), spanning different spatial and temporal scales.

Weaknesses:

I acknowledge that I am not deeply familiar with the literature on tensor-based representations for physical field modeling, so I may have missed potential limitations in that aspect of the method. From my perspective, the overall design appears reasonable and well-motivated. I have some questions about the framework which are left to the "Questions" section.

---

> ### Author Rebuttal · Authors · 2025-07-30
>
> We sincerely appreciate your thoughtful feedback and  recognition of our work. Below, we address the comments (C: comment, R: response):
>
> >C1:I acknowledge that I am not deeply familiar with the literature on tensor-based representations for physical field modeling, so I may have missed potential limitations in that aspect of the method.
>
> R1: Thank you for your comment! We would like to note that using tensors to model multidimensional physical fields is still a new research direction with little prior work. Tensor-based representations have been widely used in image processing. However, to the best of our knowledge, we are the first to apply them to generative modeling of physical fields and our extensive experiments deomstrate their effectiveness.
>
> >C2: As the number of observations decreases, is there a threshold below which the model fails to generate meaningful reconstructions due to a lack of information?
>
> R2:  Thank you for the question. Diffusion models generate data based on two main sources of information: (1) the prior distribution learned during pre-training, and (2) the likelihood distribution derived from observed data. As a generative model, a well-trained diffusion model can always produce meaningful outputs even when there are few or no observations, relying mostly on the learned prior when the likelihood information is limited.
> To illustrated that,  we plot the unconditional generation results in in Fig. 6,7,8 of Appendix F. One can see that even without observations, our method still generate meaningful trajectories.
>
>
>
>
> >C3:How does the model perform under non-uniform observation patterns, such as those common in weather-related applications, where sensors are densely located in developed regions but sparse over oceans or remote areas? This setting might challenge the assumption of homogeneity in sampling patterns.
>
> R3: Thank you for the great question! Following your suggestion, we equally divide the physical field into two regions. Assuming an observation rate of 1% at each timestep, we simulate non-uniform observation patterns by allocating **60%** of the observation points to one region and **40%** to the other. We also test a more imbalanced case, with **80%** of the observations in one region and **20%** in the other. These experiments are conducted on a single sample from each dataset. We report the reconstruction VRMSE below:
>
> | Dataset | Uniform Pattern | Non-uniform Pattern (60%/40%) | Non-uniform Pattern (80%/20%) |
> |:-------:|:----------------:|:------------------------------:|:------------------------------:|
> | Supernova Explosion ($\rho = $1%) | 0.265 | 0.271 | 0.277 |
> | Ocean Sound Speed ($\rho = $1%)   | 0.137 | 0.139 | 0.141 |
> | Active Matter ($\rho = $1%)       | 0.114 | 0.115 | 0.126 |
>
> We find that our method remains robust under non-uniform observation patterns in different datasets, highlighting its practical usefulness. We will supplement it  in the latest version.
>
> >C4:  Could you clarify the main advantages of the method compared to such approaches, particularly in terms of flexibility, generalizability, or computational efficiency?
>
> R4: We appreciate the reviewer’s insightful comment regarding the comparison with domain-specific methods such as [1], which perform field reconstruction without pretraining. Below, we highlight the advantages of our **SDIFT** framework in terms of **flexibility**, **generalizability**, and **computational efficiency**:
>
> |      **Methods**        | **Flexibility**                                                                                          | **Generalizability**                                                                                  | **Computational Efficiency**                                                                                      |
> |--------------|----------------------------------------------------------------------------------------------------------|---------------------------------------------------------------------------------------------------------|--------------------------------------------------------------------------------------------------------------------|
> | **SDIFT**    | Handles irregular and sparse observations across multiple scales during both training and testing.       | Adapts to arbitrary observation patterns during sampling.                                     | Compresses high-dimensional fields into a low-dimensional latent space. With fixed pre-trained latent functions, the likelihood model is linear, and the gradient-guided sampling is highly efficient. |
> | **Model in [1]** | Relies on predefined dynamical models (i.e., PDE/ODE model) and structured observations.                                    | Limited to low-dimensional chaotic systems with fixed observation schemes.                              | Operates in the original data space using extensive iterative gradient descent, which is costly for large systems. |
>
>
>
> - **SDIFT is able to handle irregular, off-grid observations over continuous spatiotemporal domains.** Leveraging the FTM with the universal approximation property, SDIFT flexibly represents high-dimensional fields into latent space without requiring explicit ODE/PDE formulations. Then, in latent space, SDIFT use GPSD to flexibly model temporally coherent core sequences at arbitrary timesteps. However,  [1] relies on physics-informed architectures with predefined dynamics and structured observations (e.g., periodic sampling).
>
>
>
> - **SDIFT generalizes across diverse physical systems and observation scenarios** through two key components: FTM and MPDPS. FTM constructs high-quality mappings between irregular multidimensional physical data and a lower-dimensional latent space. MPDPS effectively propagates sparse observation information across the entire core sequence, ensuring robust performance with irregular, sparse, or incomplete data. In contrast,  [1] relies on gradient-based solvers that require structured observations and degrade under irregularities.
>
>
> - **SDIFT achieves high sampling efficiency** by operating in a low-dimensional latent space. The FTM inherents a linear structure between latent factor functions and core tensors, resulting in a linear likelihood model. This design makes SDIFT scalable to large, high-dimensional systems. In contrast, [1] requires extensive iterative gradient computations over full state vectors, which becomes computationally prohibitive at scale.
>
>
> - **Regarding physical constraints**:  We agree that [1] enforces physics through explicit modeling, but this comes at the cost of flexibility and generalizability. In contrast, SDIFT implicitly captures physical regularities through FTM’s low-rank spatial structure, spatial smoothness, and GP-based temporal continuity. This design preserves flexibility while allowing the future incorporation of domain-specific priors—striking a balance between physical fidelity and adaptability to under-constrained, sparse data.

---

> > ### Comment · Reviewer_uVzD · 2025-08-07
> >
> > Thank you for your response and the additional experiments. I have one question regarding the pre-training procedure. In Section 3.1 of the paper, it is mentioned that the training data consists of regular sparse observations, which gives the impression that only sparse observations are used for pre-training. However, your response (R2) suggests that prior dynamics are also incorporated during pre-training. If prior dynamics are indeed included, then very sparse observations alone may not yield meaningful reconstructions. Could you please clarify how and when the pre-training is conducted? I would appreciate it if you could elaborate on this point.

---

> > > ### Author Response · Authors · 2025-08-07
> > > **Replying to Official Comment by Reviewer uVzD**
> > >
> > > Thank you for your feedback! We would like to clarify the pre-training phase. Assume we have a total of $B$ physical dynamics, split into a training set with $B_1$ and a test set with $B_2$ dynamics, where $B_1 + B_2 = B$ and the two sets are disjoint. To simulate realistic scenarios where sensor data is inherently sparse and irregular, we randomly mask out 80%/85% of the data in the $B_1$ set, using the remaining 20%/15% to train FTM and GPSD. During testing, we sample only 1% or 3% of the data from the $B_2$ set to reflect practical situations with highly sparse observations.
> > >
> > > We agree that very sparse observations alone in pre-training dataset  may not yield meaningful reconstructions. Therefore, we claim that the observation rate of pre-training dataset **should be much higher than the testing dataset**, so that  the learned factor functions and core sequences are meaningful.
> > >
> > >  To achieve this, we claim that solving (8) should be a well-posed problem. We can thus derive a  threshold for the observation rate (denoted as $\rho_\text{pre-training}$) of pre-training data:
> > >
> > > Assume the complete pre-trained dataset is of size $B_1 \times T \times I_1 \times I_2 \times I_3$ and the total number of learnable parameters consists of:
> > >
> > > - 1. The total parameters (denoted as $M$) for modeling the three latent  factor functions $f_{\theta_1}^1,f_{\theta_2}^2, f_{\theta_3}^3$.
> > > - 2. The size of learned core sequences $B_1 \times T \times R_1 \times R_2 \times R_3$.
> > >
> > >
> > > Theoretically, for the FTM training to remain well-posed, the following inequality must hold:
> > >
> > > $$B_1 \times T \times R_1 \times R_2 \times R_3 + M \leq \rho_\text{pre-training} \times B_1 \times T \times I_1 \times I_2 \times I_3$$
> > >
> > > Since physical dynamics exhibit strong spatiotemporal correlations, we can leverage low-rank approximation to reduce the number of parameters in the FTM(i.e., $R_i <<I_i,  \forall i$).  **As a result, our method does not require the full training dataset to generate meaningful core sequences, which enhance its flexbility. In our experiments, we find that using only  $\rho_\text{pre-training}$=15% or 20% is sufficient to produce meaningful reconstructions.**
> > >
> > > We will supplement the above disscusion in to the latest version. Thanks once again for your  constructive comments.

---

> > > > ### Comment · Reviewer_uVzD · 2025-08-08
> > > >
> > > > Thanks for your clarification and it addressed my concerns.

---

> > > > > ### Author Response · Authors · 2025-08-09
> > > > >
> > > > > Thank you for your feedback and  support！

---

### Official Review · Reviewer_dp1M · 2025-07-01

**Clarity:** 2
**Significance:** 2
**Originality:** 2
**Rating:** 3
**Confidence:** 4

**Summary:**

This paper presents SDIFT (Sequential Diffusion in Functional Tucker space), a new generative framework for reconstructing multidimensional physical dynamics from irregular and sparse spatiotemporal observations. The key innovation is the use of a functional Tucker model for latent space representation, combined with a Gaussian Process-based sequential diffusion to generate core sequences, and a Message-Passing Posterior Sampling mechanism (MPDPS) for effective conditional sequence generation. Theoretical foundation is added via a universal approximation guarantee for the functional Tucker model. The method is evaluated on three diverse real-world physical datasets (astronomical, oceanographic, and molecular), showing significant improvements in accuracy and computational efficiency over state-of-the-art baseline approaches.

**Questions:**

See weekness.

**Ethical Concerns:**

["NO or VERY MINOR ethics concerns only"]

**Final Justification:**

I still have some reservation on the novelty of this work (incremental compared with existing tensor learning methods). Hence, I maintain my original rating.

**Limitations:**

See weekness.

**Paper Formatting Concerns:**

None.

**Quality:**

3

**Strengths And Weaknesses:**

**Strengths:**

- The experimental evaluation is thorough, spanning three distinctly different domains (supernova, ocean sound speed, active matter) and comparing against a diverse set of relevant baselines. The method demonstrates consistent quantitative improvements, as detailed in Table 1 (Section 5): SDIFT with MPDPS achieves lowest VRMSE scores across datasets and observation scenarios, often with large gaps to other methods.

- The figures provide clear qualitative evidence: Figure 3 and Figure 4 demonstrate that SDIFT with MPDPS yields visually superior reconstructions at both dense and missing observation times, compared to competing approaches. In particular, Figure 4 highlights the advantage of the message-passing mechanism in interpolating missing temporal slices.

**Weaknesses:**

- The comparison with baselines is not comprehensive. Although the authors compare parameter size and runtime in Table 2, this comparison is limited to CoNFiLD only and does not include other baselines. I am particularly curious about how the proposed method performs against the other reported results.

- The reviewer is confused about the content in Section 3.1. In this section, the authors propose the Functional Tucker Model (FTM) as a universal latent representer. For Equation (6), let us consider a two-dimensional scenario and even ignore the temporal index $t$, assuming $y(i_x, i_y, t) = e^{i_x \cdot i_y}$. The reviewer would like the authors to provide finite-dimensional solutions for $f^1_{\theta_1}(i_x)$ and $f^2_{\theta_2}(i_y)$ in this example.

- Building upon the previous concern, for the case $y(i_x, i_y, t) = e^{i_x \cdot i_y}$, the reviewer argues that only an approximate solution can be obtained under finite-dimensional settings. This raises a new issue: in the proof of universal approximation, the authors must account for the case where $R^k \to \infty$. Therefore, the reviewer is confused by the last step of Equation (24) on page 15.

- There is a typographical error in the formula on page 14, Equation (18). In the $B_2$ component, $f_{r_1}^{A_1}$ should be $f_{r_1}^{k}$.

- Minor issue: references [28] and [38] refer to the same publication.

---

> ### Author Rebuttal · Authors · 2025-07-30
>
> We thank the reviewer for the careful review! We address the comments below:( C: comment; R: response)
>
>
> >C1: The comparison with baselines is not comprehensive. Although the authors compare parameter size and runtime in Table 2, this comparison is limited to CoNFiLD only and does not include other baselines. I am particularly curious about how the proposed method performs against the other reported results.
>
> R1: Thank you for the comment. We compared our method only with CoNFiLD because both are generative approaches. The parameter sizes and runtimes for the other methods are reported below.
>
> |     |SDIFT (ours) |CoNFiLD|NONFAT | DEMOTE | LRTFR | Senseiver|
> |:-----:|:-----:|:-----:|:-----:|:-----:|:-----:|:-----:|
> | Param. size on Supernova Explosion |26M|23M| N/A| 6.5M |4.2M|9.3M|
> | Samp. time and VRMSE  on Supernova Explosion (1%) |**2.23s(0.283$\pm$0.026)**|31.3s(0.561$\pm$0.082)| $>30$min(1.229$\pm$0.127)| $>30$min(1.285$\pm$0.102) |31.2s(0.558$\pm$0.044)|0.83s(0.446±0.041)|
> | Samp. time and VRMSE on Supernova Explosion  (3%)|**5.43s(0.272$\pm$0.025)**| 44.0s(0.427$\pm$0.037)| $>30$min(1.197$\pm$0.204) | $>30$min(1.213$\pm$0.217) |35.4s(0.429$\pm$0.043)|1.5s(0.349±0.023)|
> | Param. size on Ocean Sound Speed |15M|10M| N/A| 5.8M |3.2M|8.5M|
> | Samp. time  and VRMSE on Ocean Sound Speed  (1%)|**0.84s(0.146$\pm$0.046)**|15.9s(0.201$\pm$0.034)| $>30$min(0.402$\pm$0.090) | $>30$min(0.358$\pm$0.127) |27.5s(0.345$\pm$0.036)|0.036s(0.264±0.037)|
> | Samp. time and VRMSE on Ocean Sound Speed  (3%)| **0.89s(0.108$\pm$0.043)**|20.3s(0.145$\pm$0.012)| $>30$min(0.330$\pm$0.101) | $>30$min(0.314$\pm$0.086) |28.0s(0.217$\pm$0.066)|0.04s(0.2005±0.031)|
> | Param. size on on ActiveMatter  |12M|10M| N/A | 4.7M | 3.2M|8.2M
> | Samp. time and VRMSE on Active Matter  (1%)|**1.31s(0.215$\pm$0.068)**| 27.6s(0.529$\pm$0.087)| $>30$min(0.921$\pm$0.457) | $>30$min(0.950$\pm$0.486) |26.9s(0.302$\pm$0.104) |0.14s(0.345±0.094)
> | Samp. time and VRMSE on Active Matter  (3%)|**1.42s(0.156$\pm$0.046)**| 31.5s(0.5075$\pm$0.830)|$>30$min(0.867$\pm$0.413)| $>30$min(0.871$\pm$0.497) | 27.3s(0.258$\pm$0.022) |0.18s(0.264±0.076)
>
>
> The parameter sizes of different models vary because they are different types of models and  we fine-tune each model individually to achieve its best performance, taking into account its unique properties.
>
> We observe that tensor-based methods (NONFAT, DEMOTE, and LRTFR) require significantly more time to produce reconstructions. This is because they directly minimize a loss function that enforces consistency between predicted and observed values, without any pretraining phase. As a result, these methods need extensive iterations to converge and are much slower than pretrained models. Among them, LRTFR is considerably faster than NONFAT and DEMOTE, as the latter two are not scalable to large tensors.
>
>
>
> Senseiver is the most efficient model in terms of runtime, as it feeds the indices to be queried into the network and performs the forward pass only once to produce the reconstructions. However, it does not explicitly model the temporal evolution of the physical field. As a result, its performance is inferior to ours and it fails when no observations are available.
>
>
> >C2: The reviewer is confused about the content in Section 3.1.
>
> R2: Thank you for your question. In your example, $y(i\_1, i\_2) = e^{i\_1 \cdot i\_2}$, **we can construct a finite-dimensional FTM solution that exactly represents $y$.** That is $y(i\_1,i\_2)= \text{vec}(\mathbf{W})^{\text{T}}(f\_{\theta\_1}^1(i\_1) \otimes f\_{\theta\_2}^2(i\_2))$, where $\mathbf{W}$ being a scaler 1 and $f\_{\theta\_1}^1(\cdot)=e^{i\_1}: \mathbb{R}^1 \to \mathbb{R}^1$,$f\_{\theta\_2}^2(\cdot)=e^{i\_2}: \mathbb{R}^1 \to \mathbb{R}^1$    being scaler functions. Then the FTM model becomes $1\cdot(f\_{\theta\_1}^1(i\_1) \cdot f\_{\theta\_2}^2(i\_2))$. Since both latent functions are parameterized by neural networks, they can accurately approximate the exponential functions involved.
>
>
> For illustration, let's consider a finite 2-order tensor(matrix) $\mathbf{Y}$ with size $3 \times 3$ with each entry $y(i\_1,i\_2)=e^{i\_1,i\_2}$. Then  $\mathbf{Y} = $
>
> |     |     |     |
> |:-----:|:-----:|:-----:|
> | y₁₁ | y₁₂ | y₁₃ |
> | y₂₁ | y₂₂ | y₂₃ |
> | y₃₁ | y₃₂ | y₃₃ |
>
> =
> |     |     |     |
> |:-----:|:-----:|:-----:|
> | $e^1$ | $e^2$ | $e^3$ |
> | $e^2$ | $e^4$ | $e^6$ |
> | $e^3$ | $e^6$ | $e^9$ |
>
>
> FTM decomposes the tensor $\mathbf{Y}$ into a Tucker core and mode-wise latent functions.
> Assume $R\_1 = R\_2 = 1$, then the Tucker core is a learnable matrix $\mathbf{W} \in \mathbb{R}^{1}$, and the latent functions are defined as:
> - Mode-1: $f\_{\theta\_1}^1(\cdot): \mathbb{R}^1 \to \mathbb{R}^1$
> - Mode-2: $f\_{\theta\_2}^2(\cdot): \mathbb{R}^1 \to \mathbb{R}^1$
>
> Here, $f\_{\theta\_1}^1(\cdot)$ and $f\_{\theta\_2}^2(\cdot)$ are neural networks that learn positional embeddings and are parameterized by $\theta\_1$ and $\theta\_2$.
>
> To approximate the value at index $(1, 1)$, FTM computes: $  \text{vec}(\mathbf{W})^\top \left( f\_{\theta\_1}^1(1) \otimes f\_{\theta\_2}^2(1) \right) \approx y\_{11} = e^{1}$.
>
> Other entries in $\mathbf{Y}$ are computed similarly.
>
> FTM minimizes the reconstruction error between $\mathbf{Y}$ and the model output, learning optimal values for $\mathbf{W}$, $f\_{\theta\_1}^1(\cdot)$, and $f\_{\theta\_2}^2(\cdot)$.  It still works even when some entries in $\mathbf{Y}$ are missing.
> After training,  $\left( \mathbf{W}, f\_{\theta\_1}^1(\cdot), f\_{\theta\_2}^2(\cdot) \right)$ can be used to represent the original tensor $\mathbf{Y}$. Therefore, FTM serves as an encoder.
>
>
>
>
> >C3: Confusion on the case $R\_k \to \infty$ and  last step of Equation (24).
>
>
> R3:  Thank you for the question. We would like to clarify that the universal approximation proof is based on **sufficiently large but finite** $R\_k$, rather than taking $R\_k \to \infty$.
>
> As shown in [A1], the Tucker model is a universal representer: for sufficiently large $\{R\_k\}$—up to $R\_k = I\_k$, where $I\_k$ is the size of mode $k$—any finite-dimensional tensor can be exactly represented via Tucker decomposition.
>
> For example, consider the above case where $\mathbf{Y} \in \mathbb{R}^{3\times 3}$, $I\_1 = I\_2 = 3$. If we choose $R\_1 = R\_2 = 3$, i.e., $\mathbf{W} \in \mathbb{R}^{3 \times 3}$, then the model
>
> $$\text{vec}(\mathbf{W})^\top \left( f\_{\theta\_1}^1(\cdot) \otimes f\_{\theta\_2}^2(\cdot) \right)$$
>
> can accurately represent $\mathbf{Y}$.
> Therefore, $R\_k$ is a finite and bounded value, and the expression in Eq. (24) consists of finite summations.
>
> Moreover, natural physical fields typically exhibit low-rank structure, as discussed in [20,21], implying that the effective $R_k$ values are often small in practice.
>
>
>
> [A1] N. D. Sidiropoulos, L. De Lathauwer, X. Fu, K. Huang, E. E. Papalexakis, and C. Faloutsos, “Tensor decomposition for signal processing and machine learning,” IEEE Trans. Signal Process., vol. 65, no. 13, pp. 3551–3582, Jul. 2017.
>
> >C4:There is a typographical error in the formula on page 14, Equation (18)
>
> R4: Thank you for pointing out this error. We will fix it in the latest version.
>
> >C5:Minor issue: references [28] and [38] refer to the same publication.
>
> R5: Thank you for pointing out this issue. We will address it in the latest version.

---

> > ### Comment · Reviewer_dp1M · 2025-08-06
> >
> > **Comment 0:**  After re-reading the manuscript, I have identified additional issues.
> >
> > First, the tensor decomposition–based approach employed in the manuscript has already been extensively applied in dynamical modeling and prediction (see, e.g., [1], [2]). The authors should more clearly position their method relative to these existing works and explicitly articulate the novel contributions.
> >
> > Second, the proposed Gaussian Process–based Sequential Diffusion model warrants a comparison with standard autoregressive approaches. I recommend providing either direct experimental results or at least a discussion to clarify the advantages and potential limitations of the proposed method relative to autoregressive baselines.
> >
> > ---
> >
> > **Comment 1:**  The authors’ clarification has satisfactorily addressed my earlier concern regarding runtime performance.
> >
> > ---
> >
> > **Comment 2:**  Regarding the request for a finite-dimensional construction of $e^{i_x \cdot i_y}$, the example
> > $$
> > y(i_x, i_y) = \vec{w}^\top \big( f_1(i_x) \otimes f_2(i_y) \big),
> > $$
> > where $\vec{w}$ is the scalar $1$, $f_1(i_x) = e^{i_x}$, and $f_2(i_y) = e^{i_y}$, yields $y = e^{i_x + i_y}$ rather than $e^{i_x \cdot i_y}$. This is because the tensor product of separable scalar functions produces an additive exponent, not the multiplicative interaction term present in $e^{i_x \cdot i_y}$.
> >
> > ---
> >
> > **Comment 3:**  Concerning the claim that
> > $$
> > \vec{w}^\top \big( f(\cdot) \otimes g(\cdot) \big)
> > $$
> > can exactly represent $e^{i_x \cdot i_y}$, I note that $e^{xy}$ does not admit an exact finite-dimensional separable expansion of the form
> > $$
> > e^{xy} = \sum_{n_1 = 1}^N \sum_{n_2 = 1}^N a_{n_1 n_2} f_{n_1}^{(1)}(x) f_{n_2}^{(2)}(y), \quad N < \infty,
> > $$
> > valid for all $x, y \in \mathbb{R}$.
> >
> > **Proof:**  Assume, for contradiction, that such a finite representation exists.
> >
> > Let $\{x_i\}_ {i = 1}^M$ and $\{y_j\}_ {j = 1}^M$ be any two sets of distinct real numbers. Define
> > $$
> > \mathbf{M} = \left[e^{x_i y_j}\right]_ {i,j = 1}^M = \sum_ {n_1 = 1}^N \sum_ {n_2 = 1}^N a_{n_1 n_2} f_{n_1}^{(1)}(x_i) f_{n_2}^{(2)}(y_j).
> > $$
> > Each term $f_ {n_1}^{(1)}(x_i) f_ {n_2}^{(2)}(y_j)$ is a rank‑1 matrix over $(i,j)$, so $\mathbf{M}$ is the sum of at most $N^2$ rank‑1 matrices, implying
> > $$
> > \mathrm{rank}(\mathbf{M}) \leq N^2.
> > $$
> > However, it is known [3, p.9, Eq. (3.1)] that for any choice of distinct $\{x_i\}$ and $\{y_j\}$, the matrix $\mathbf{M} = [e^ {x_i y_j}]$ has nonzero determinant and is therefore full rank:
> > $$
> > \mathrm{rank}(\mathbf{M}) = M.
> > $$
> > Choosing $M > N^2$ yields a contradiction:
> > $$
> > M = \mathrm{rank}(\mathbf{M}) \leq N^2 < M.
> > $$
> > Thus, no such finite separable representation exists.
> >
> > ---
> >
> > **References:**
> > [1] Chen, Yuanhong, Yifan Lin, Xiang Sun, Chunxin Yuan, and Zhen Gao. "Tensor decomposition-based neural operator with dynamic mode decomposition for parameterized time-dependent problems." *Journal of Computational Physics* 533 (2025): 113996.
> >
> > [2] Chen, Xinyu, Yixian Chen, Nicolas Saunier, and Lijun Sun. "Scalable low-rank tensor learning for spatiotemporal traffic data imputation." *Transportation Research Part C: Emerging Technologies* 129 (2021): 103226.
> >
> > [3] Karlin, Samuel, and William J. Studden. *Tchebycheff Systems: With Applications in Analysis and Statistics*. Interscience Publishers, 1966.

---

> > > ### Author Response · Authors · 2025-08-08
> > >
> > > Dear Reviewer dp1M,
> > >
> > > Could we kindly know if the responses have addressed your concerns and if further explanations or clarifications are needed? Your time and efforts in evaluating our work are appreciated greatly！

---

> > > > ### Comment · Reviewer_dp1M · 2025-08-08
> > > > **Thanks for your response**
> > > >
> > > > Thanks for your clarifications. Although I still have some reservation on the novelty of this work (incremental compared with existing tensor learning methods), I do not have any further technical questions.

---

> ### Author Response · Authors · 2025-08-07
> **Replying to Official Comment by Reviewer dp1M(1)**
>
> Thank you for your feedback! We address the comments below:( C: comment; R: response)
>
>
> R0: We sincerely thank the reviewer for the insightful comments. Regarding the relationship between our method and prior tensor-decomposition work (e.g.,  Chen *et al.* [1] and  Chen *et al.* [2]), we agree that tensor factorization has proved powerful for spatiotemporal modeling.  **Our contribution, however, targets a different problem setting: generating full-field evolutions of physical dynamics from sparse, noisy, and continuously indexed observations.** The differences between our method with these existing tensor-based methods are:
>
> - (i) Existing tensor-based approaches are typically trained to deterministically reconstruct or extrapolate a single trajectory on a regular grid;  They often struggle to perform reliably under extremely sparse observations. However, **SDIFT learns a full generative distribution over sparse, irregular spatiotemporal coordinates** and it can always produce meaningful off-grid results.
> - (ii) Instead of directly factorizing the observed tensor like [1][2], we use a **Functional Tucker latent space**, where coordinates are embedded via continuous functions alongside a learnable, time-varying core. This framework defines a flexible manifold that GPSD can effectively model.
> - (iii) The time evolutions in Chen *et al.* [1] are based on a **linear dynamic mode decomposition** assumption, while Chen *et al.* [2] relies on **decoupled mode factors**. These assumptions limit their ability to represent complex dynamics. In contrast, our **GP-driven sequential diffusion** framework effectively captures **nonlinear dynamics** and provides **predictive uncertainty**.
>
> - (iv) SDIFT enables conditional generations of dynamics **at arbitrary continuous time steps** with the proposed MPDPS mechanism whereas Chen *et al.* [1] and Chen *et al.* [2] do not.
>
>
>
> To better clarify our positioning, we will reference both works and include a dedicated subsection in the related work discussing differences in task formulation, sampling strategies, and uncertainty modeling.
>
>
> **Concerning autoregressive (AR) alternatives**, we acknowledge their popularity in image and video generation (i.e.,  MaskGIT[B1], diffusion forcing [B2]) and have already included **Senseiver**—an AR attention model—as a key baseline. SDIFT reduces its VRMSE by 22 % on the same sparsity level, and and Senseiver fails entirely in the observation setting 2. The  gap between our method and AR approaches arises from four key factors:
> - (i) **Irregular, continuous timestamps** force AR models to discretise Δt, which introduces interpolation bias.
> - (ii) **Extremely sparse spatial observations** make pixel-space AR training infeasible.
> - (iii) **Sampling flexibility:** SDIFT solves an ODE to “jump” directly to any target time, while AR models must generate frames sequentially.
> - (iv) **Inference strategy:** our message-passing posterior sampling integrates limited observations into the full sequence in a single pass, something AR models cannot consistently achieve. Thus, we claim it's non-trival to apply the AR directly in our problem setting.
>
> Therefore, we argue that directly applying existing autoregressive (AR) models designed for computer vision (CV) or natural language processing (NLP) to our problem setting is non-trivial. In response to the reviewer’s request, we will note in the Limitations section that, **when observations are dense and uniformly sampled**, a simple AR backbone may indeed offer a more efficient solution. We will also expand the discussion to consider future work on hybrid AR–diffusion frameworks.
>
> Thank you again for your valuable feedback!
>
> [B1] Chang, Huiwen, et al. "Maskgit: Masked generative image transformer." Proceedings of the IEEE/CVF conference on computer vision and pattern recognition. 2022.
>
> [B2] Chen, Boyuan, et al. "Diffusion forcing: Next-token prediction meets full-sequence diffusion." Advances in Neural Information Processing Systems 37 (2024): 24081-24125.

---

> ### Author Response · Authors · 2025-08-07
> **Replying to Official Comment by Reviewer dp1M(2)**
>
> R1: Thanks for the feedback!
>
> R2: Thank you for pointing this out. We sincerely apologize for our misunderstanding. We claim that there is no finite-dimensional separable expansion  solution  to  represent $e^{i \times j}$ with FTM.
>
> R3: Thank you for your thoughtful comment and clear proof! We agree that $e^{xy}$ does not admit an exact finite-dimensional separable expansion in the FTM form. **However, this does not contradict our proposed Universal Approximation Property (UAP) theorem.**
>
>
> Our UAP states that FTM can approximate any continuous function to arbitrary precision (i.e., for any $\epsilon > 0$, where $\epsilon$ is controlled by the choice of bounded values $R_1$ and $R_2$). This **does not** imply that FTM always yields an exact finite-dimensional separable representation for every continuous function. Instead, it focuses on **characterizing the asymptotic approximation error**.
> **Typically, exact separable representations are not required in practical applications, as approximate forms are sufficient to achieve reliable performance, as demonstrated in our experiments.**
>
>
>
>
> Additionally, we would like to argue that if  the dimension $M$ is not infinite (i.e., bounded), we can always choose a sufficiently large $N$ (e.g., $N = M$) such that FTM can represent the matrix $\mathbf{M}$ exactly (i.e., $\epsilon = 0$), though the resulting form is not exactly separable expansion.
>
> We sincerely appreciate your careful checking and constructive feedback and we will make this point explicit in the main text!

---

> ### Author Response · Authors · 2025-08-08
>
> Thank you for your feedback. However, we beg to differ with the reviewer’s comment that our work is “incremental compared with existing tensor learning methods.”
>
>  We would like to clarify that the **Functional Tucker Model (FTM)** is used solely  as a latent representation mapping in our framework. The directly using of functional tensor is **only a fraction of main contributions of our method**. Beyond that, we provide a theoretical proof of the **universal approximation property** of FTM in this context firstly, to support its applicability.
>
> What's more, our primary contributions lie in the design of **GPSD** and **MPDPS** on the side of **generative methods**, which together enable to infer entire physical fields at **arbitrary coordinates and timesteps** in a **generative manner**. Such setting and paradigm are practically significant,  technically underexplored, and  **fundamentally different compared with any existing tensor decomposition methods**.

---

### Official Review · Reviewer_wHTh · 2025-07-04

**Clarity:** 3
**Significance:** 3
**Originality:** 4
**Rating:** 5
**Confidence:** 4

**Summary:**

The authors present Sequential Diffusion in Functional Tucker Space (SDIFT). They deal with the problem of posterior sampling of a continuous-time real-vector valued sequence (like solutions of a PDE). Motivated by latent space diffusion, they do posterior sampling given partial observations of the whole continuous time signal by doing diffusion modeling and posterior sampling in the latent space of a functional tucker tensor decomposition. In the work they deal with two main challenges (1) define the latent space according to the tucker decomposition in order to handle sparse/off-time-grid observations and (2) perform inference using likewise incomplete observations to generate a full continuous time field.

**Questions:**

- in section 3 124, for the diffusion prior pre-training phase, what's the definition of homogenous in "B batches of homogeneous dynamics" mean?

- in section 3.1, line 166, in equation 8 , do you need a coefficient in front of the TV term in practice? If so, how do you choose it?

- section 3.2, line 192, equation 9, there is a denoiser-matching objective to train the diffusion model for fixed W.  Have you tried training (8) and (9) simultaneously rather than in pieces. Any +/-'s to report?

- does temporally augmented Unet just mean that the Unet gets the noised $(W_t)^{s}$ (W for data time t at diffusion time s) as well the times $t=[t_1,...t_M]$ themselves? If not, what does it mean?

- *Main question*: What is proved that the specific message passing algorithm introduced, which uses $\nabla_{W^s_{\setminus
t_\ell}} \log p(O_{t_\ell} | W^s_{\setminus
t_\ell})$ ? I get intuitively that we are passing some info from the observed to the rest, but abstracting a bit, how can I see more easily that updating the whole $W_t$ according to these gradients produces a sample of $W_t^{\text{all}} | W_t^{\text{obs}}$ ?

**Ethical Concerns:**

["NO or VERY MINOR ethics concerns only"]

**Final Justification:**

My positive opinion of the work remains unchanged.

**Limitations:**

Yes.

**Paper Formatting Concerns:**

Yes.

**Quality:**

3

**Strengths And Weaknesses:**

Strengths

- precise mathematical formalization of the problem at hand
- moving between typical VAE latent space to something more structured that is used in the applied math / numerical community, but mixing with strengths of ML + diffusion. The time-varying core tensors $W_{t_k}$ of the tucker decomposition are taken as the latent representation so that for data spatial index $i_k$ (very loosely) we have$y_{i,t} = w_t f^k_\theta(i_k)$ for learnable $f^k_\theta$ and learnable $w_t$ for mode $k$ . The cores $w_t$ share info across the $K$ modes' latent functions
- the latent space $W(t)$ is still at time varying function, so they use functional diffusion models to model it (functional diffusion model has GP noise)
- the diffusion pre-training and posterior inference algorithms are separated into two clean pieces (where the first learning piece itself has two parts: learn the decomposition, learn the diffusion)
- novel message passing algorithm to perform the inference
- convincing experiments on real scientific tasks (1) supernova explosion temperature evolution (2) pacific ocean sound speed (3) active particles in simulated fluid with comprehensive description/links for more details to all data details.

Weaknesses

See questions.

---

> ### Author Rebuttal · Authors · 2025-07-30
>
> Thank you for your strong support!  Here are our responses. C: comments; R: response.
>
> >C1:in section 3 124, for the diffusion prior pre-training phase, what's the definition of homogenous in "B batches of homogeneous dynamics" mean?
>
> R1: Thank you for your question. In generative modeling, people assume that the training data is extracted from a specific distribution. Here, we also assume the training data is extracted from a specific distribution of  dynamics. The homogeneous dynamics refer to the same type of dynamics, such as the same PDE or ODE, but with different initial conditions.
>
> >C2:in section 3.1, line 166, in equation 8 , do you need a coefficient in front of the TV term in practice? If so, how do you choose it?
>
> R2: Good point! We add a coefficient in front of the TV term. We have done cross-validation experiments to choose the coefficient and we find that $10^{-6}, 10^{-7}, 10^{-8}$ all work. We will clarify it in the latest version.
>
>
> >C3:section 3.2, line 192, equation 9, there is a denoiser-matching objective to train the diffusion model for fixed W. Have you tried training (8) and (9) simultaneously rather than in pieces. Any +/-'s to report?
>
> R3:Good question. Theoretically, (8) and (9) can be trained simultaneously. However, the objectives of FTM and GPSD differ significantly:
>
> - **FTM** aims to learn a high‑fidelity mapping between the data domain and latent space, minimizing reconstruction error.
> - **GPSD** aims to model the distribution of latent‑space dynamics over time.
>
> Jointly optimizing these distinct objectives leads to slower training and reduced stability.
>
> To address this, we adopt a two‑stage training pipeline：first training the FTM, then freezing it and training the GPSD. This strategy is widely used in the computer vision literature [A1], which has been shown to yield superior generation quality, faster sampling, and more stable convergence compared to end‑to‑end training.
>
>
>
>
> [A1] Rombach, Robin, et al. "High-resolution image synthesis with latent diffusion models." Proceedings of the IEEE/CVF conference on computer vision and pattern recognition. 2022.
>
> >C4:does temporally augmented Unet just mean that the Unet gets the noised
>  ($\mathbf{W}\_{t}^{s}$ for data time t at diffusion time s) as well the times $t=[t\_1,\cdots,t\_M]$ themselves? If not, what does it mean?
>
>  R4: The answer is yes. Apart from that, we also  use Conv1D layers to capture temporal correlations among $M$ frames. The detailed archetecture is provided in Appendix B.
>
>
> >C5:Main question: What is proved that the specific message passing algorithm introduced, which uses $\nabla\_{W\_{\setminus t\_l}^s}\log p(O\_{t\_l}|W\_{\setminus t\_l}^s)$ . I get intuitively that we are passing some info from the observed to the rest, but abstracting a bit, how can I see more easily that updating the whole $\mathbf{W}\_t$ according to these gradients produces a sample of $\mathbf{W}\_t^{\text{all}}|\mathbf{W}\_t^{\text{obs}}$.
>
>
> R5: Thank you for your question! We want to highlight that the primary goal of MPDPS is to exploit the temporal continuity of the core sequence to **smoothly propagate guidance from limited observations** across the entire sequence.
>
> For clarity, consider the setup in Fig. 2, where we aim to generate the core tensors at three target timesteps: $t\_l$, $t'$, and $t\_{l+1}$, denoted as **$\mathcal{W}\_{t\_l}$**, **$\mathcal{W}\_{t'}$**, and **$\mathcal{W}\_{t\_{l+1}}$**, respectively. In this example, observations are only available at $t\_l$ and $t\_{l+1}$ (observation timesteps),denoted as $\mathcal{O}\_{t\_l}$ and $\mathcal{O}\_{t\_{l+1}}$. Standard DPS methods cannot provide gradient guidance for generating the intermediate core **$\mathcal{W}\_{t'}$**, due to the lack of direct observations at $t'$.
>
> **MPDPS addresses this limitation by propagate the information of $\mathcal{O}\_{t\_l}$ to {**$\mathcal{W}\_{t\_l}, \mathcal{W}\_{t'},\mathcal{W}\_{t\_{l+1}}$**} and the information of $\mathcal{O}\_{t\_{l+1}}$ to {**$\mathcal{W}\_{t\_l}, \mathcal{W}\_{t'},\mathcal{W}\_{t\_{l+1}}$**},  which means each observation set will contribute to the generation of all target sequences.**
>
>
>
> Without loss of generality, we take $\mathcal{O}\_{t\_l}$ as an example. At diffusion step $s$, we use $\mathcal{O}\_{t\_l}$ *twice* to obtain gradient guidance for the three cores:
>
>
> - **First use to compute gradient for $\mathcal{W}\_{t\_l}^s$**:  This follows standard DPS. We directly compute the likelihood gradient with respect to $\mathcal{W}\_{t\_l}^s$ as  $\nabla\_{\mathcal{W}\_{t\_l}^s} \log p(\mathcal{O}\_{t\_l} | \mathcal{W}\_{t\_l}^s)$, as illustrated by the green& red arrows at $t\_l$ in Fig.2(a).
>
> - **Second use to compute gradients for $\mathcal{W}\_{t'}^s$ and $\mathcal{W}\_{t\_{l+1}}^s$**:
> It includes three steps:
>     1.  **Compute denoised cores at $t'$ and $t\_{l+1}$**: we first use $\mathcal{W}\_{t'}^s$ and $\mathcal{W}\_{t\_{l+1}}^s$ to estimate the corrsponding denoised cores $\hat{\mathcal{W}}\_{t'}^0 = D\_{\theta}(\mathcal{W}\_{t'}^s)$ and $\hat{\mathcal{W}}\_{t\_{l+1}}^0 = D\_{\theta}(\mathcal{W}\_{t+1}^s)$ via the pretrained denoiser $D\_{\theta}(\cdot)$.
>
>     2. **Estimate pseudo cores at $t\_l$ via GPR**:  With the temporal continuity assumption, we use Gaussian Process Regression (GPR) to estimate the pseudo denoised core  $\hat{\mathcal{W}}\_{t\_l}^0 = GPR (\hat{\mathcal{W}}\_{t'}^0, \hat{\mathcal{W}}\_{t\_{l+1}}^0) = GPR(D\_{\theta}(\mathcal{W}\_{t'}^s), D\_{\theta}(\mathcal{W}\_{t+1}^s))$ , which can be viewed as a regression-based prediction using two denoised cores.
>     3. **Gradient computation**: With pseudo core $\hat{\mathcal{W}}\_{t\_l}^0$ $\quad$, we compute an additional likelihood of $\mathcal{O}\_{t\_l}$. Then we can then compute the gradient of the likelihood respect to ${\mathcal{W}}\_{t'}^s$ and ${\mathcal{W}}\_{t\_{l+1}}^s$, as they directly determine $\hat{\mathcal{W}}\_{t\_l}^0$ via $D\_{\theta}$ and GPR. We further derive a **closed-form solution** in Appendix C for gradients with respect to $\mathcal{W}\_{t'}^s$ and $\mathcal{W}\_{t\_{l+1}}^s$ (i.e., $\nabla\_{W\_{\setminus t\_l}^s}\log p(O\_{t\_l}|W\_{\setminus t\_l}^s)$ where  $W\_{\setminus t\_l}^s=${$\mathcal{W}\_{t'}^s,\mathcal{W}\_{t\_{l+1}}^s$}), which can be efficiently computed. The results are shown in Eq.(10)(11).
>     The process is illustrated by the green arrows pointing to $\mathcal{O}\_{t\_l}$ in Fig. 2(b).
>
>
>
> Similarly, information from $\mathcal{O}\_{t\_{l+1}}$ is propagated to **$\mathcal{W}\_{t\_l}$** 、**$\mathcal{W}\_{t'}$**, and **$\mathcal{W}\_{t\_{l+1}}$** in the same manner, and we sum the gradients from all observations to obtain the final gradient guidance for each core, as shown in Eq. (12).
>
> In this way, each observation contributes to the generation of all target timesteps. In other words, every target core—regardless of whether it has a corresponding observation—receives information from all observed timesteps. To sample the entire core sequence, we simply add the gradients from all observations to the posterior, as shown in Eq. (12). This mechanism encourages smooth and coherent generation across the entire sequence.
>
>
> We hope this explanation clarifies how MPDPS updates the entire core sequence. We will include this in the latest version.

---

### Comment · Area_Chair_sKJf · 2025-08-08
**Diverging opinions in Reviews**

Dear Reviewers,

thx for engaging in discussions with the authors. Your opinions are diverging, so it would help my work if you could give a qualified statement during the AC-reviewer discussion period.

If you need further information from the authors, can you try to use the remaining time for clarifying questions or answering their comment?

Thx, AC

---

### Decision · Program_Chairs · 2025-09-17

**Decision:**

Accept (poster)

**Comment:**

This submission received two clearly positive reviews (wHTh, uVzD, both rating 5) and two more skeptical, borderline reviews (dp1M, BZPv, both rating 3). The supportive reviewers praise the method’s practical importance, soundness, and strong empirical results across diverse physical systems. They highlight the novelty of combining a Functional Tucker latent space with Gaussian process–driven sequential diffusion and a message-passing posterior sampling mechanism. Both positive reviewers found the paper well-motivated, technically solid, and convincingly validated.

The more critical reviewers raise concerns primarily about (i) the level of novelty relative to existing tensor-based spatiotemporal methods, (ii) positioning with respect to prior work, (iii) limited discussion of broader ML significance, and (iv) clarity of presentation and notation. While dp1M questions whether the approach is incremental, they acknowledge its technical correctness and strong results, and explicitly state they would not strongly oppose acceptance. BZPv also notes clarity and framing issues, but concedes the method is novel and effective for the stated problem, and would not object to acceptance. These two reviewers are more senior.

The authors provided detailed clarifications and additional experiments. They addressed concerns on baseline coverage, positioned their work relative to prior tensor decomposition methods, and explained the specific advantages over autoregressive approaches and domain-specific PDE solvers. They also added robustness tests for non-uniform observation patterns and clarified the pretraining process. While the skeptical reviewers maintained their original ratings, they indicated no remaining technical objections after rebuttal.

I read the paper and reached conclusions consistent with the reviewers:
 - Strengths: Strong empirical performance on challenging, diverse datasets; technically sound method; interesting integration of latent functional tensor decomposition with generative modeling.
 - Limitations: Novelty is moderate, given prior tensor and GP work; the write-up and notation could be streamlined for clarity; broader ML relevance is not strongly argued.

This is an exact borderline case: the paper is well-executed and solves an important, realistic problem, but with moderate novelty and clarity concerns. Given the strong empirical results, the fact that no reviewer is strongly negative, and that the technical correctness is not in dispute, I lean toward giving the benefit of the doubt.